# Ribogenesis boosts controlled by HEATR1-MYC interplay promote transition into brain tumour growth

Laura R Diaz[1,7], Jon Gil-Ranedo [1,7], Karolina J Jaworek [1,2], Nsikan Nsek [1], Joao Pinheiro Marques [1], Eleni Costa[1], David A Hilton[3], Hubert Bieluczyk[1], Oliver Warrington[1,5], C Oliver Hanemann [1], Matthias E Futschik [4,6], Torsten Bossing [1] & Claudia S Barros [1✉]

## Abstract

Cell commitment to tumourigenesis and the onset of uncontrolled growth are critical determinants in cancer development but the early events directing tumour initiating cell (TIC) fate remain unclear. We reveal a single-cell transcriptome profile of brain TICs transitioning into tumour growth using the *brain tumour* (*brat*) neural stem cell-based *Drosophila* model. Prominent changes in metabolic and proteostasis-associated processes including ribogenesis are identified. Increased ribogenesis is a known cell adaptation in established tumours. Here we propose that brain TICs boost ribogenesis prior to tumour growth. In *brat*-deficient TICs, we show that this dramatic change is mediated by upregulated *HEAT-Repeat Containing 1* (*HEATR1*) to promote ribosomal RNA generation, TIC enlargement and onset of overgrowth. High *HEATR1* expression correlates with poor glioma patient survival and patient-derived glioblastoma stem cells rely on HEATR1 for enhanced ribogenesis and tumourigenic potential. Finally, we show that HEATR1 binds the master growth regulator MYC, promotes its nucleolar localisation and appears required for MYC-driven ribogenesis, suggesting a mechanism co-opted in ribogenesis reprogramming during early brain TIC development.

**Key words** Neural Stem Cells; Brain Tumourigenesis; Ribogenesis; HEATR1; MYC
**Subject Categories** Cancer; Signal Transduction; Translation & Protein Quality

## Introduction

Cancer is an outcome of events whereby normal cells acquire driver mutations leading to their transformation, endowing adaptive advantages and uncontrolled growth. The acquisition of unrestricted mitotic ability, ignoring proliferation control signals, is a hallmark of all malignant cells (Hahn and Weinberg, 2002; Puisieux et al, 2018). (Re)-initiation and fuel of many cancers rely on a subpopulation with self-renewal properties analogous to organ stem or progenitor cells termed cancer stem cells (CSCs) or tumour initiating cells (TICs) (Ayob and Ramasamy, 2018; Clarke et al, 2006). TICs undergo genetic and epigenetic reprogramming to promote tumour development (Batlle and Clevers, 2017; Puisieux et al, 2018). Yet, the complexity of mammalian tumour models renders the characterisation of early tumourigenesis challenging and major gaps remain in our understanding of the early events following oncogenic insults and leading to tumour growth.

We made use of a genetically defined *Drosophila* neural stem cell (NSC)-based tumour model in which TICs and their cells of origin are characterised and can be traced from the point of transformation. The *Drosophila* larval brain harbours stereotyped NSC lineages generating neurons and glia of the adult brain, which serve as a model to stem cell and cancer research (Hakes and Brand, 2019; Homem and Knoblich, 2012). A subset of NSCs, termed type II, behave similarly to mammalian counterparts, dividing asymmetrically and giving rise to intermediate neural progenitors (INPs). INPs have restricted stem potential undergoing 5–6 asymmetric divisions to self-renew and generate ganglion mother cells (GMCs) that divide once producing neurons or glia (Bello et al, 2008; Boone and Doe, 2008; Bowman et al, 2008). Type II NSCs express the transcription factor Deadpan (Dpn) promoting self-renewal and are identifiable by Ets transcription factor Pointed (PntP1) expression and absence of proneural Asense (Ase) that is present in other NSCs (Brand et al, 1993; Zhu et al, 2011). Newly born INPs are immature (iINPs) and lack Ase and Dpn protein expression, which they acquire after a maturation process lasting

¹Peninsula Medical School, Faculty of Health, John Bull Building, University of Plymouth, PL6 8BU Plymouth, UK. ²School of Biological Sciences, Bangor University, LL57 2UW Bangor, UK. ³Department of Cellular and Anatomical Pathology, University Hospitals Plymouth, PL6 8DH Plymouth, UK. ⁴School of Biomedical Sciences, Faculty of Health, Derriford Research Facility, University of Plymouth, PL6 8BU Plymouth, UK. ⁵Present address: Wellcome Centre for Human Neuroimaging, UCL Queen Square Institute of Neurology, University College London, WC1N 3AR London, UK. ⁶Present address: Center for Innovative Biomedicine and Biotechnology (CIBB), University of Coimbra, 3004-504 Coimbra, Portugal. ⁷These authors contributed equally: Laura R Diaz, Jon Gil-Ranedo. ✉E-mail: claudia.barros@plymouth.ac.uk

4–6 h before initiating mitosis. Brat is a Tripartite Motif (TRIM)-NHL (NCL1, HT2a and LIN41) tumour suppressor protein asymmetrically segregated from NSCs into INPs (Betschinger et al, 2006; Lee et al, 2006) where it binds the mRNA and post-transcriptionally represses *dpn* as well as *zelda* (zld), another transcription factor required for self-renewal (Reichardt et al, 2018). Upon *brat* loss, iINPs fail to acquire identity and transform into TICs leading to tumour growth. The *brat* brain tumour model allows for the study of stepwise acquisition of tumour stem cell properties, tumour growth and progression. *brat*-deficient iINPs cannot mature. Instead, they constitutively express the NSC markers Dpn and Zld, but not Ase, and undergo a transient cell cycle delay. After 24–48 h, they begin overproliferating leading to malignant growth (Bowman et al, 2008). Restoring *brat* expression 24 h after *brat* depletion prevents the abnormal INPs from forming a tumour. Yet, after 48 h it can no longer prevent tumour growth, indicating that at this point they are irreversibly committed to tumourigenesis (Bonnay et al, 2020). In contrast to normal NSCs, *brat* tumour fragments or FACS-sorted TICs re-form tumours when transplanted into healthy hosts. These tumours have been grown for years when serially transplanted and metastasise leading to host death (Caussinus and Gonzalez, 2005; Landskron et al, 2018; Laurenson et al, 2012).

The human Brat orthologue, TRIM3, is also a brain tumour suppressor demonstrated by repression of patient-derived glioblastoma (GBM) stem cells (GSCs) tumourigenic potential and growth of intracranial xenografted GBM cells in mice (Chen et al, 2014; Liu et al, 2014). GBM is the most common malignant (WHO grade IV) brain tumour in adults (Stupp et al, 2005). Deletion mapping analysis identified *TRIM3* loss in 25% of GBMs (Boulay et al, 2009) and cancer database probing reveals *TRIM3* deletions in 24% of grade II–IV gliomas including 20–22% in GBM. TRIM3 protein is reduced in GBM and absent in GSCs (Chen et al, 2014; Liu et al, 2014; Mukherjee et al, 2016). Similar to Brat's, TRIM3's NHL domain is also required for its growth suppressive properties shown in glioma cells (Arama et al, 2000; Komori et al, 2014; Liu et al, 2014). NHL domains mediate protein–protein and protein-RNA interactions. Brat's NHL is necessary for binding to *dpn* and *zld* mRNAs. TRIM3 has been suggested to also bind RNA (Williams, 2021) yet potential mRNA targets remain unknown. Brat and TRIM3 can attenuate Notch signalling at least in part via suppressing nuclear transport of Notch intracellular domain (NICD) in a process dependent on the Importin complex (Mukherjee et al, 2016); in addition, both inhibit the expression of the oncogene MYC in *Drosophila* brains and GBM tumourspheres, respectively (Betschinger et al, 2006; Chen et al, 2014; Song and Lu, 2011; Zaytseva et al, 2020).

To examine early tumourigenesis, we interrogated *brat*-deficient brain TICs when these show molecular markers of transformation but have not yet begun to overgrow (Bowman et al, 2008). We performed single-cell transcriptome profiling on TICs and compared with control INP counterparts individually isolated directly from live brains. The data highlight changes in conserved metabolic and proteostasis processes including ribogenesis. Ribogenesis involves orchestration of proteins and nucleic acids to respond to multiple inputs needed for protein synthesis and homeostasis. Its hyperactivation is an accepted adaptation of cancer cells to uncontrolled proliferation that relies on increased protein synthesis. Yet, how ribogenesis determines early TIC development

remains unclear (Bastide and David, 2018; Pelletier et al, 2018). Our results demonstrate that *brat*-deficient brain TICs boost ribogenesis ahead of tumour growth in a fashion dependent on high HEATR1 expression. Patient-derived GSCs also require HEATR1 for enhanced ribogenesis, tumourigenic potential and growth. We further show that HEATR1 binds the master ribogenesis and cell growth regulator MYC and promotes its localisation to ribogenesis sites, suggesting a mode of action to enhance ribogenesis during early TIC development stimulating the transition into brain tumour growth.

## Results

### *brat* brain TICs display enhanced proteostasis and metabolic transcriptomic signatures before overgrowth onset

To identify mechanisms driving early TIC development, we took advantage of the well-characterised *brat* brain tumour model in which TICs are known (*brat* INPs) (Bowman et al, 2008) and performed a small-scale single-cell analysis of their transcriptome from live brains. By combining *pnt-gal4* with *UAS-CD8-GFP* transgenic lines in control and *brat* backgrounds, cell membranes of type II NSC lineages harbouring healthy and tumour initiating INPs (*brat* TICs) were specifically labelled. *brat* TICs and control (iINPs) were individually manually harvested from brains at 24 h after larval hatching (ALH), an early timepoint when *brat* INPs show molecular markers indicating their transformation into TICs but have not yet started overproliferating to form a tumour (Bowman et al, 2008) (Fig. 1A, left panels; n = 3 *brat* TICs, n = 3 control iINPs). To minimize potential differences due to spatial positioning, cells were removed from type II NSC lineages in the dorsal-median anterior region of brain lobes (DM1 and DM2 NSC lineages). Only INPs adjacent to their larger NSC precursor, i.e. recently born INPs (Bayraktar and Doe, 2013), were removed, maximizing retrieval of iINPs from control brains. Using our protocol (Barros and Bossing, 2022; Bossing et al, 2012), cDNA from each individual cell was readily obtained. PCR on each single cell cDNA confirmed that *brat* TICs and control iINPs show no mature INP marker *ase* expression as expected. Expression profiling was obtained by comparing transcriptomes (Fig. 1A, right panels; see Methods). Principal Component Analysis (PCA) addressing sample variance confirms that *brat* TICs and controls cluster in respective groups (Fig. 1B). Using LPE analysis (Murie and Nadon, 2008), 358 transcripts were identified as differentially expressed (FDR < 0.1; Fig. 1C and Dataset EV1). The data show orthology conservation, with most genes with highly conserved human (70%) and mouse (67%) orthologues and only 11% with no mammalian counterpart (Fig. 1D and Dataset EV1). For quality control, we examined the expression of a subset of identified candidates (33) using independent single-cell cDNA samples (n = 3 *brat* TICs, n = 3 controls) isolated following the same procedure as for the transcriptome analysis and real-time quantitative PCRs (RT-qPCRs). Differential expression was confirmed for all candidates tested (Fig. 1E).

Although the *Drosophila* and human brain share numerous properties, they show major species-specific differences. The use of *Drosophila* brain tumour models needs therefore to focus on genetic

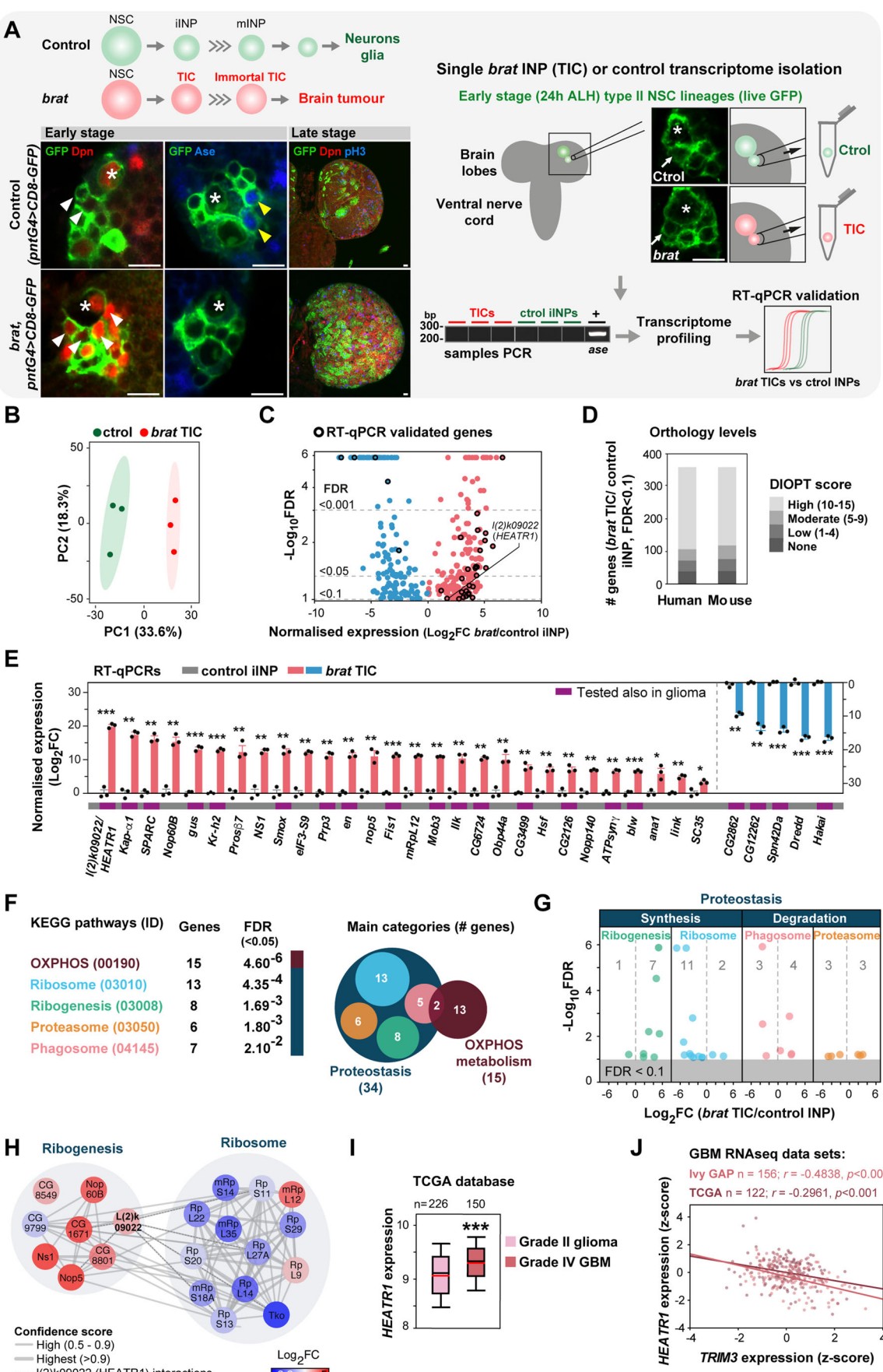

Figure 1. Transcriptomics of *brat* brain TICs before tumour growth onset.

(A) Workflow: *brat* TICs and control iINPs expressing *CD8-GFP* were individually isolated from 24 h ALH live brains when *brat* INPs transform into TICs aberrantly expressing Dpn (red; white arrowheads) but do not yet overproliferate. Unlike control iINPs, TICs do not mature (Ase⁻, blue; yellow arrowheads) and lead to tumour growth (Bowman et al, 2008) (left panels). cDNA was obtained from each single cell harvested and PCRs confirmed *ase* absence. Transcriptomes were compared on whole-genome microarrays (biological replicates: n = 3 TICs; n = 3 ctrl iINPs). Data validated by RT-qPCRs (right panels). Asterisk: NSCs. Arrows: INPs. Scale bars: 10 μm. (B) PCA plot with 95% prediction ellipses of *brat* TICs and controls. (C) Volcano plot of differentially expressed genes (fold change, FC; FDR < 0.1). (D) Human and mouse orthology level (single best matches) of differentially expressed genes. (E) RT-qPCRs of a subset of identified genes (biological replicates: n = 3 *brat* TICs versus 3 control iINPs; error bars: s.e.m.; unpaired two-tailed t-tests followed by Holm correction). (F, G) Overrepresented KEGG pathways (dataset FDR < 0.1; F). Identified genes within proteostasis pathways (Number, FC expression, FDR; G). (H) Protein–protein interaction network of identified ribogenesis and ribosome candidate molecules by the KEGG pathway analysis shown in (F). Node colour: gene expression levels (Log₂FC); Line thickness: confidence scores. L(2)k09022 (*Drosophila* HEATR1) interactions: dashed lines. (I) Grade IV gliomas express higher levels of *HEATR1* than grade II gliomas. Box plots represent 25th and 75th percentiles, central black bands indicate medians, central red bands specify means, whiskers indicate 10th and 90th percentiles. Unpaired two-tailed t-test. Biological replicates: n = 226 (grade II glioma); 150 (GBM). (J) *TRIM3* and *HEATR1* expression inversely correlate in GBM (r, Pearson's correlation coefficient). Biological replicates: 156 (GBM; Ivy Gap), 122 (GBM; TCGA). Data information: ***p ≤ 0.001; **p ≤ 0.01; *p ≤ 0.05. Source data are available online for this figure.

and phenotypic features that can be addressed in the fruit fly (Read, 2011) and discoveries probed in mammalian systems. In the past decades, studies using *Drosophila* enhanced our understanding of aspects of the tumourigenic potential of brain cell types and of tumour development (Hakes and Brand, 2019; Homem and Knoblich, 2012). The *brat* model provides the opportunity to examine very early stages of tumorigenesis. Gene orthology and functional conservation exists between *brat* and *TRIM3*, and *TRIM3* deletions are observed in 20% of all gliomas (Fig. EV1A) (Boulay et al, 2009; Chen et al, 2014; Liu et al, 2014; Mukherjee et al, 2016). These features prompted us to investigate human orthologues of genes identified in our data in glioma. First, we ascertained *TRIM3* downregulation in high (GBM, *IDH* wild-type) and lower grade (grade II diffuse astrocytoma, DA, *IDH* mutant) glioma samples, and TRIM3 absence in patient-derived GSCs consistent with literature (Fig. EV1B and C) (Boulay et al, 2009; Chen et al, 2014; Liu et al, 2014; Mukherjee et al, 2016). Next, we performed expression analysis of our candidate genes between tumour or GSCs and non-tumour brain samples. We found that 60% of candidates show differential expression in GBM (12/20), 60% in grade II DAs (12/20) and 68% in GSCs (13/19) (Fig. EV1D). Interestingly, comparing our human gene dataset (DIOPT 5-15) to that of MacLeod and colleagues following CRISPR knockdown screens in 10 GSC lines (MacLeod et al, 2019) by gene set enrichment analysis (GSEA) reveals a highly significant enrichment in genes reported to contribute to GSC fitness (Fig. EV1E and Dataset EV2).

To gain more insight of events potentially directing early tumourigenesis, we performed KEGG pathway enrichment data analysis. In addition to oxidative phosphorylation (OXPHOS) metabolism, the most enriched pathways relate to protein home-ostasis: synthesis (ribogenesis and ribosome) and degradation (phagosome and proteasome functions) (Fig. 1F,G and Dataset EV3). OXPHOS metabolism and proteostasis pathways are also most overrepresented in the orthologue human data (Fig. EV1F and Dataset EV3).

An altered proteostasis network in cancerous cells to meet elevated growth rates is widely recognised (Bastola et al, 2018; Harper and Bennett, 2016). Yet, our data suggest an imbalanced proteome in TICs prior to tumourous growth. Among the identified proteostasis KEGG pathways (Fig. 1F), candidates associated with ribosome and ribogenesis form the largest network and include *l(2)k09022*, the orthologue of human *HEATR1*, which we refer herein also as *HEATR1* (Fig. 1H and Dataset EV1). HEATR1 was reported in one previous study to be overexpressed in GBM compared to non-tumour brain tissues (Wu et al, 2014) but

its role in brain tumourigenesis is unknown. We confirmed HEATR1 overexpression in GBM (Figs. EV1D, EV2A,B). Inter-rogating TGCA, REMBRANDT and CGGA databases reveals that *HEATR1* expression in glioma inversely correlates with patient survival (Fig. EV2C). Database analysis also indicates *HEATR1* upregulation in GBM compared to grade II gliomas (Fig. 1I). Accordingly, we observe higher HEATR1 levels in GBM versus grade II DAs (Fig. EV2A). Using Ivy GAP, which reports on gene expression within putative CSC versus non-CSC clusters in GBM, reveals stronger *HEATR1* expression in the former. We also find *HEATR1* upregulated in GSCs versus control brain samples, and its protein overexpressed in GSCs versus human foetal NSCs (Figs. EV1D and EV2D,E). Finally, since we identified *HEATR1* over-expressed in *brat* TICs, we explored TGCA and Ivy GAP data to investigate a potential correlation between the expression of the respective human orthologues. *HEATR1* and *TRIM3* levels inversely correlate in GBM samples and within defined intra-tumour regions, with lower *TRIM3* and higher *HEATR1* expression observed in regions with higher percentage of tumour cells (Figs. 1J and EV2F).

Collectively, the results show that our single-cell transcriptome analysis generated quality data exposing conserved genes poten-tially involved in early brain tumourigenesis. Together with metabolic changes, the data highlight adaptation of proteostasis processes and in particular ribogenesis. To explore ribogenesis in early brain TIC development, we next expanded our analysis of the identified ribogenesis-associated HEATR1.

## *brat*-deficient brain TIC transition into tumourous growth requires HEATR1

We sought to address the impact of *HEATR1* inhibition on early *brat* tumourigenesis. Targeted *brat*^RNAi in type II NSC lineages elicits tumourigenesis reproducing the *brat* null phenotype as the self-renewal marker Dpn is not repressed in iINPs (Bowman et al, 2008; Reichardt et al, 2018). Simultaneous *CD8-GFP* expression identifies type II NSC lineages in *brat*-deficient tumours and controls. At 24 h ALH, an early timepoint when *brat* INPs are transformed into TICs but have not yet started to overproliferate (Bowman et al, 2008), *HEATR1*^RNAi expression has no effect in the proliferation of control cells with self-renewal ability (GFP⁺, Dpn⁺, phospho-Histone H3, pH3⁺) nor in *brat*-deficient cells (Fig. 2A–E; see Appendix Fig. S1A for mitotic index quantifications). To address cell fate and transformation, GFP, Dpn and Ase markers

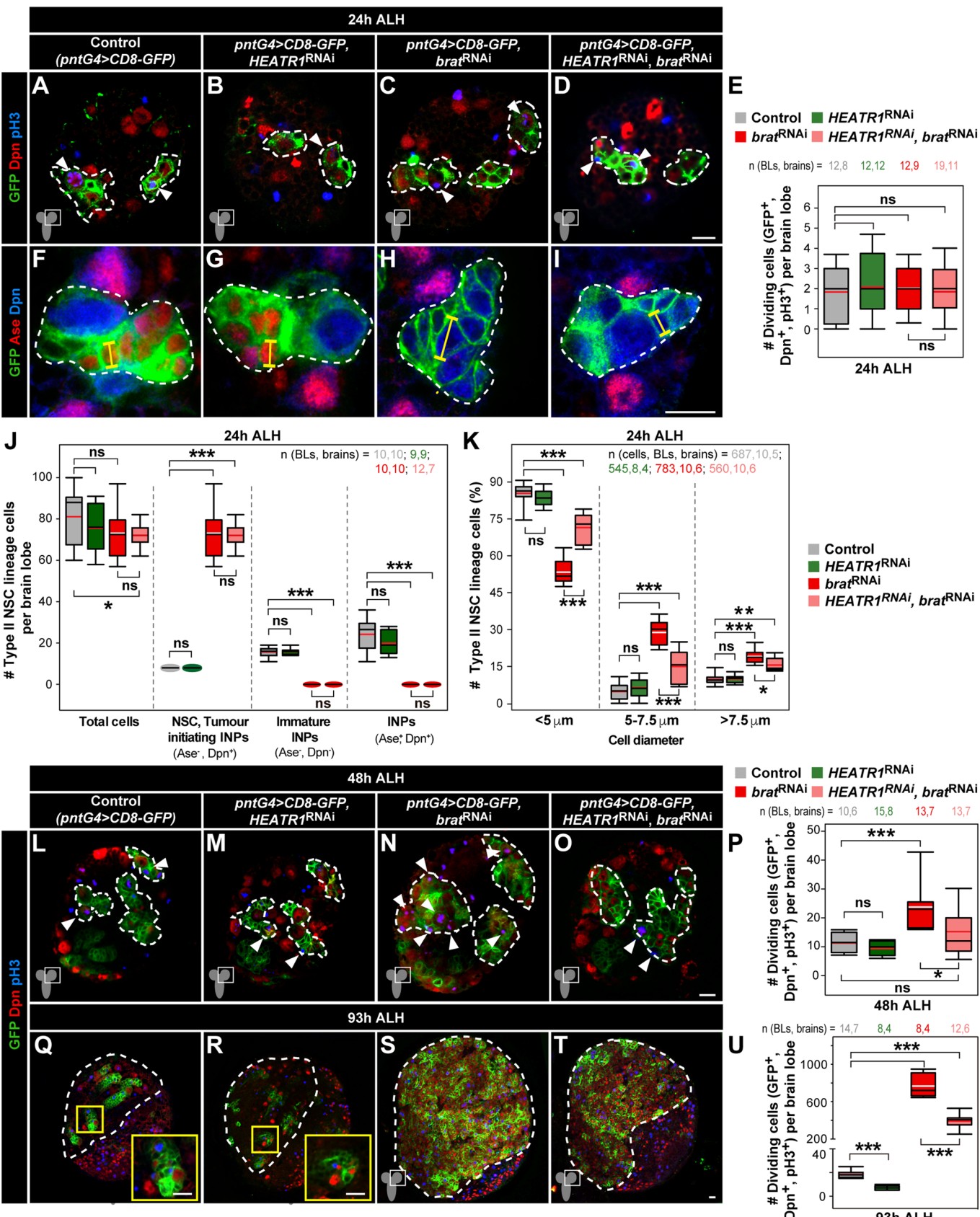

**Figure 2. HEATR1 promotes *brat*-deficient brain TIC enlargement and tumour overgrowth onset.**

(A–E) Immunostainings of GFP, Dpn and pH3 in type II NSC lineages expressing *CD8-GFP* (control) and *HEATR1*[RNAi], *brat*[RNAi] (tumour) or both *HEATR1*[RNAi], *brat*[RNAi] (*HEATR1*-deficient tumour) at 24 h ALH (A–D). Quantification of mitotic cells (GFP$^+$Dpn$^+$pH3$^+$) per brain lobe (E; biological replicates: 12–19). Unpaired two-tailed t-tests. (F–K) Immunostainings of GFP, Ase and Dpn in type II NSC lineages of control, *HEATR1*-deficient only, tumour and *HEATR1*-deficient tumour at 24 h ALH (F–I). Yellow lines: maximum cell diameter. Quantification of cell types in type II NSC lineages per brain lobe (J; biological replicates: 9–12). Quantification of cell sizes in type II NSC lineages per brain lobe (K; biological replicates: 8–10). Unpaired two-tailed t-tests. (L–U) Immunostainings of GFP, Dpn and pH3 in type II NSC lineages of control, *HEATR1*-deficient only, tumour and *HEATR1*-deficient tumour at 48 h ALH (L–O) and 93 h ALH (Q–T). Quantification of mitotic cells (GFP$^+$Dpn$^+$pH3$^+$) per brain lobe at 48 h ALH (P; biological replicates: 10–15) and 93 h ALH (U; biological replicates: 8–14). Insets: higher magnification. Unpaired two-tailed t-tests, except P (Mann–Whitney test, *brat*[RNAi] versus control). Data information: Scale bars: 10 µm. Dashed lines: type II NSC lineages (A–D, F–I, L–O), central brain region (Q–T). Arrowheads: pH3$^+$ cells. Brain Lobes, BL. Genotypes are indicated. Box plots represent 25th and 75th percentiles, central black bands indicate medians, central red or white bands specify means, whiskers indicate 10th and 90th percentiles. ***$p \leq 0.001$; **$p \leq 0.01$; *$p \leq 0.05$; $p > 0.05$, ns (non-significant). See mitotic cell index quantifications (Appendix Fig. S1A–C). Source data are available online for this figure.

were used. Ase is not present in type II NSCs and only becomes expressed in INPs when these start to mature. Controls show a larger type II NSC (Dpn$^+$, Ase$^-$) and smaller INPs, which are first immature (Dpn$^-$, Ase$^-$) and become mature (Dpn$^+$, Ase$^+$) before generating further committed progeny, whereas *brat*-deficient type II NSC lineages are composed of a NSC and TICs (Dpn$^+$, Ase$^-$) (Fig. 2F,H,J) (Bello et al, 2008; Boone and Doe, 2008; Bowman et al, 2008). Upon *HEATR1*[RNAi], the number of NSCs, immature and mature INPs in control lineages is unchanged indicating no cell fate alterations, and *brat*-deficient TICs remain Dpn$^+$, Ase$^-$ (Fig. 2G,I,J). In addition to NSC-like properties such as expressing Dpn, *brat* tumour cells are larger compared to non-tumour counterparts (Betschinger et al, 2006; Bowman et al, 2008). At 24 h ALH, NSCs measure 7–8 µm while INP size is 4–6 µm in non-tumour controls (Ding et al, 2016; Egger et al, 2008; Gil-Ranedo et al, 2019). We found that already at this early stage there is a reduced number of smaller cells (<5 µm) and increased number of larger cells in *brat* depleted lineages, demonstrating enlarged TIC size prior to overproliferation onset (Fig. 2F,H,K). Knock-down of *HEATR1* in non-tumour controls has no effect in the size of any cell type (Fig. 2F,G,K). Yet, in *brat*-deficient lineages, it leads to a decrease in larger cells and concomitant increase in the smaller group, indicating that *brat*-deficient TIC size is rescued to levels closer to that of control INPs (Fig. 2H,I,K).

We next analysed *HEATR1* inhibition at 48 h ALH, a developmental stage when *brat* TICs started to overproliferate to form a tumour and will grow indefinitely as demonstrated via serially transplant paradigms (Bowman et al, 2008; Caussinus and Gonzalez, 2005; Landskron et al, 2018). At this stage, *HEATR1* knockdown in control type II NSC lineages has no impact on proliferation but, strikingly, in *brat*-deficient tumour cells it prevents overgrowth onset, with proliferation levels per brain lobe detected similar to those in controls (Fig. 2L–P; see Appendix Fig. S1B for mitotic indexes). Finally, we examined the effect of HEATR1 inhibition in *brat* tumours at late stages (93 h ALH). *HEATR1*[RNAi] expression strongly prevents brain tumour growth with proliferation levels reducing approximately by half and respective mitotic index back to levels similar to controls (Fig. 2S–U; Appendix Fig. S1C). At this late stage, depleting *HEATR1* inhibits proliferation in control lineages indicating it is also required in normal neural cells although this effect is not significant at earlier stages (Fig. 2Q,R,U and Appendix Fig. S1C; compare to Fig. 2A–E,L–P and Appendix Fig. S1A,B). TUNEL assays reveal that cell death levels are not affected upon *HEATR1* depletion (Fig. EV3A–E). Together, the results indicate that early *brat*-deficient brain TICs require HEATR1 for enlargement and

overgrowth onset while proliferation of controls cells at the same stage is not significantly affected.

## GBM cell proliferation and tumourigenic potential depend on HEATR1

HEATR1's overexpression in patient-derived GSCs and its requirement for *brat*-deficient brain TIC development led us to hypothesize it promotes GSC growth enhancing tumourigenesis. Endoribonuclease-prepared small interfering RNAs (esiRNA) and lentivirus-delivered short hairpin RNA (shRNA) against *HEATR1* knock it down efficiently in GBM (U87MG, U251MG) and GSCs (GSC-5 and GSC-8) derived from independent tumours (Appendix Fig. S2A,B). We found that targeting *HEATR1* decreases GBM and GSC proliferation as measured by immunostaining with the cell cycle marker Ki67 and S-phase labelling via 5-ethynyl-2'-deoxyuridine (EdU) incorporation (Fig. 3A–L). To evaluate whether HEATR1 contributes to GSC tumourigenic potential, we performed anchorage-independent growth soft agar colony assays. GSC growth from single cells into tumourspheres within agar matrices correlates with tumourigenic potential in vivo and prevents spontaneous cell aggregation (Gordon et al, 2018). HEATR1 depletion in GSCs abrogates tumoursphere formation with a dramatic reduction in number and size of colonies (Fig. 3M–T). Finally, we evaluated the impact on cell death following *HEATR1* inhibition. Similar to results in *brat*-deficient brain tumours compared to controls brains, no changes are detected in GBM cell apoptosis as seen by TUNEL assays, whereas only 5–10% increase in cell death is seen via Annexin-5 and propidium iodide labelling in one of the two GSC lines examined (Fig. EV3F,G). We conclude that HEATR1 is required in GSCs for their tumourigenic capacity and proliferation, functions also observed in vivo in *brat*-deficient tumours.

## *brat*-deficient brain TICs and GSCs rely on HEATR1 to enhance ribogenesis

Our single-cell transcriptome data indicate protein homeostasis at the forefront of molecular adaptations in early brain TIC development. Core to proteostasis is the control of ribogenesis and protein synthesis (Harper and Bennett, 2016). The primary sites of ribogenesis, nucleoli, are dynamic organelles. Mammalian nucleoli are tripartite with functionally distinct subcompartments. Transcription of rDNA by Pol I occurs at the interface between fibrillar centres (FCs) and dense fibrillar centres (DFCs). rRNA processing mediated by the small-subunit processome including

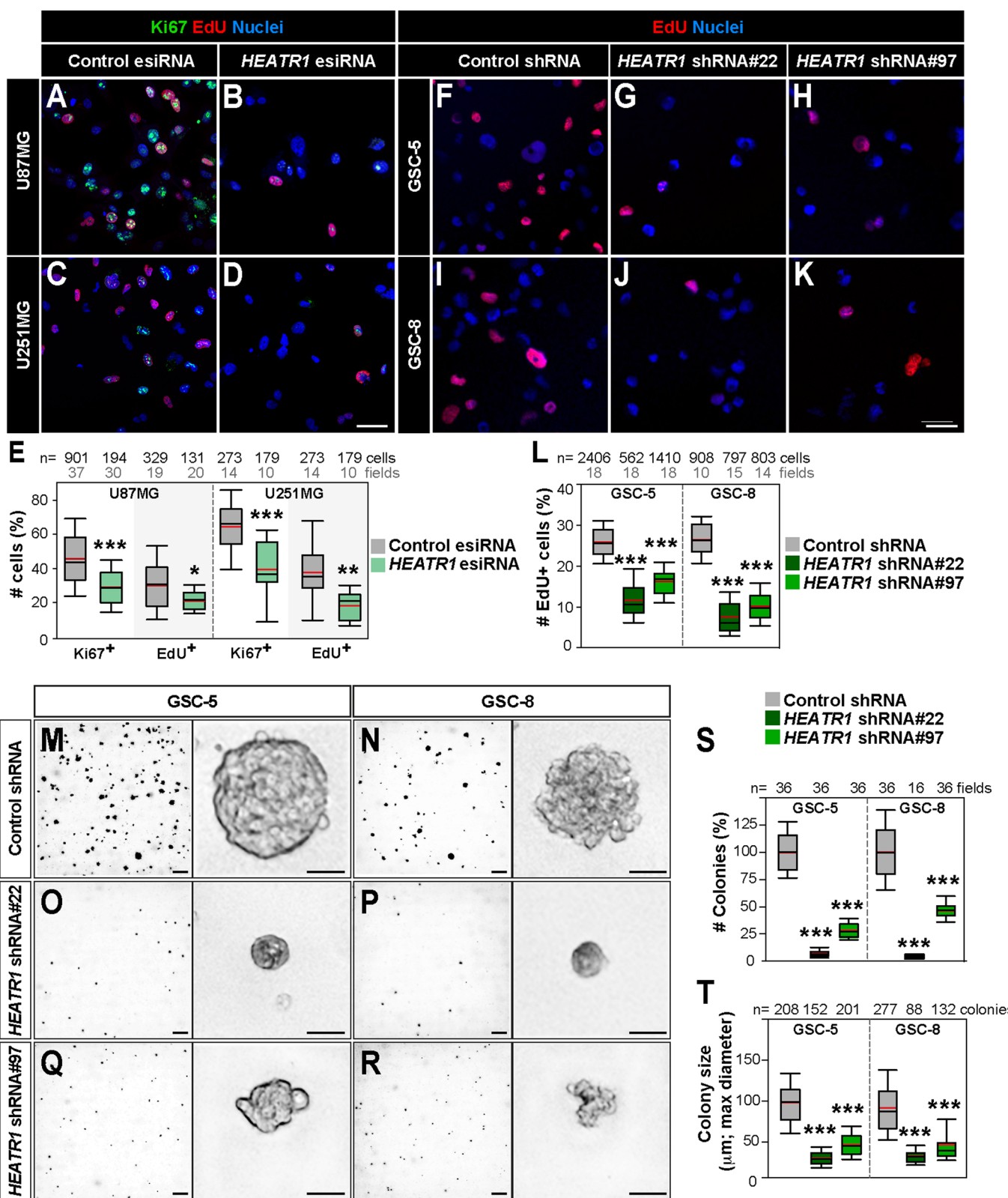

Fibrillarin (FBL) takes place at the DFCs, whereas late rRNA maturation occurs in granular components (GCs) before export of the subunits to the cytoplasm for final maturation (McStay, 2016). While *Drosophila* nucleoli are ultrastructurally less organised, they present several conserved mammalian nucleolar components including FBL (Knibiehler et al, 1982; Orihara-Ono et al, 2005). In both mammals and *Drosophila*, nucleoli size and architecture are dependent on active transcription and alterations in these reflect

**Figure 3. GBM cell proliferation and stemness potential require HEATR1.**

(A–E) Immunostainings of Ki67 and EdU-labelling (10 μM, 1 h) in GBM cells (U87MG, U251MG) 48 h post transfection (hpt) with *HEATR1*-esiRNA (B, D) or control *GFP*-esiRNA (A, C). Nuclei (DAPI). Ki67+ and Edu+ cells quantification: % of DAPI+ cells, unpaired two-tailed t-tests; E; 30–37 (U87MG Ki67+), 19–20 (U87MG Edu+), 10–14 (U251MG Ki67+) and 10–14 (U251MG Ki67+) cell images (fields) from 3 independent cell cultures (biological replicates). (F–L) EdU-labelling (20 μM, 1 h) in GSC-5 and GSC-8 168 h post-infection (hpi) with *HEATR1*-shRNAs (G, H, J, K) or control shRNAs (F, I). Nuclei (DAPI). Edu+ GSCs quantification: % of DAPI+ cells, unpaired two-tailed t-tests; L; 18 fields per condition (GSC-5) and 10–15 fields (GSC-8) from 3 biological replicates. (M–T) Soft agar colony formation analysis of GSC-5 and GSC-8 168 hpi with *HEATR1*-shRNAs (O–R) or control shRNAs (M, N). Colony number (%) quantification (S): 36 fields per condition (GSC-5) and 16–36 fields (GSC-8) from 3 biological replicates. Colony size quantification (T): 152–208 (GSC-5) and 88–277 (GSC-8) colonies from 3 biological replicates. Mann–Whitney tests. Scale bars: 50 μm except in (M–R) left panels (400 μm). Data information: Box plots represent 25th and 75th percentiles, central black bands indicate medians, central red bands specify means, whiskers indicate 10th and 90th percentiles. ***$p \leq 0.001$; **$p \leq 0.01$; *$p \leq 0.05$. Source data are available online for this figure.

changes in ribogenesis (Grewal et al, 2005; Kressler et al, 2017; Nemeth and Grummt, 2018). We sought to address HEATR1's potential role in ribogenesis sites of brain TICs in vivo. Within grown *brat* brain tumours, cells present larger nucleoli (Betschinger et al, 2006). Using immunolabelling with FBL antibodies, we demonstrate that *brat*-deficient TICs show enlarged nucleoli compared with control INPs already at 24 h ALH (Fig. 4A,A',C,C',E). This was surprising as TICs at this stage do not overproliferate. Larger nucleoli reflect increased ribogenesis and is a hallmark of many tumour types (Orsolic et al, 2016). While targeted *HEATR1* inhibition in control type II NSC lineages leads to a marginal reduction in the size of nucleoli of INPs, it completely abrogates enlarged nucleoli of *brat*-depleted TICs to levels even slightly lower than those of controls (Fig. 4B,B',D,D',E). Controls using a non-targeting RNAi strain (*cherry*RNai) and genetic background comparable to that of the RNAi lines used do not significantly alter nucleoli size (Fig. 4A,A',E and Appendix Fig. S3A–B'; see Methods). In agreement with the findings, *HEATR1* inhibition results in reduced levels of nascent RNA in *brat*-deficient TICs as measured by 5-Ethynyl Uridine (EU) incorporation (Fig. 4F–H). Since most cell's transcription is from active rDNA loci (McStay and Grummt, 2008), the reduction is likely in nascent rRNA transcribed by Pol I.

We next investigated HEATR1's role in ribogenesis of GSCs and GBM cells. We demonstrate that HEATR1 localises predominantly in the nucleoli of GSCs, GBM and lower grade glioma tissue (Fig. EV4A–J). We then examined whether its inhibition affects GSC nucleolar functional domains. GSCs depleted of HEATR1 show smaller nucleoli, an effect more prominent in FC and DFC compared to GC domains, labelled with antibodies against Upstream Binding 4 Factor (UBF), FBL and Nucleophosmin (NPM1), respectively (Fig. 4I–P). Yet, in most cells, the sub-components aberrantly distribute to nucleoli periphery with UBF and FBL often forming strong accumulations (Fig. 4I–N,Q). These resemble the so-called nucleolar caps described upon halting transcription and nucleolar stress such as upon treatment with Actinomycin D (ActD) (McStay, 2016). Consistent with impaired rDNA transcription, *HEATR1* depletion also decreases Pol I accumulation in nucleoli and re-distribution to nucleoli periphery (Fig. 4R–V). Interestingly, a concomitant decrease in UBF levels is observed but no changes in total FBL, NMP1 or Pol I levels (Fig. 4W). Analogous effects are observed in GBM cells (Appendix Fig. S4A–I).

In line with the above findings, we found that *HEATR1* inhibition leads to decreased nascent RNA levels, measured by EU incorporation (Fig. 5A–B',E) indicating decreased rRNA synthesis in GBM cells (McStay and Grummt, 2008). Indeed, blocking ribogenesis by treatment with low dosage of ActD that specifically inhibits rRNA

polymerase (Pol) I transcription but not Pol II or III activity (Boulon et al, 2010) results in dramatic reduction in EU incorporation in nuclei of GBM cells comparable to that upon *HEATR1* inhibition (Fig. 5C–E), and abolishes tumourigenic potential in GSCs as measured via soft agar assays (Appendix Fig. S5A–F, compare to Fig. 3M–T). *HEATR1* depleted cells also present decreased *47S* precursor rRNA (*47S* pre-rRNA) and of *18S*, *5.8S* and *28S* rRNAs resulting from *47S* pre-rRNA processing, as shown by relative expression levels and by copy number per cell (Fig. 5F–H). These rRNA subunits, together with *5S* rRNA transcribed by RNA Pol III, constitute the nucleic acid backbone and catalytic activity of ribosomes (Campbell and White, 2014).

*HEATR1* inhibition in osteosarcoma U2OS cells was reported to impair ribogenesis and trigger the RPL5/RPL11-MDM2-p53 ribogenesis stress checkpoint pathway resulting in accumulation of p53 tumour suppressor levels and cell cycle arrest (Turi et al, 2018). In GBM cells and GSCs we observed no significant accumulation of p53 (Appendix Fig. S6A,B). The lack of p53 increase upon *HEATR1* inhibition is not due to GBM cells being unable to activate RPL5/RPL11-MDM2-p53 signalling as low ActD dosage, which leads to the pathway activation (Holmberg Olausson et al, 2012), results in robust p53 accumulation (Appendix Fig. S6C). HEATR1's modes of action may therefore be cell context-dependent. Finally, since ribogenesis including rRNA production is rate limiting for protein synthesis (Harper and Bennett, 2016), we examined *HEATR1* depletion's impact in the latter. We observed a reduction of protein translation in GBM cells demonstrated by decreased O-propargyl-puromycin (OPP) incorporation (Fig. 5I–M).

Collectively, the data indicate that *brat*-deficient brain TICs boost ribogenesis before entering into tumourous growth, a process that requires HEATR1. HEATR1's role in ribogenesis is also seen in GSCs and GBM cells.

## HEATR1 binds to MYC and regulates its nucleolar localisation

A key stimulator of ribogenesis and cell growth is the transcription factor MYC (also known as c-Myc) (Duffy et al, 2021). Indeed, genes with ribosome and nucleolar roles are part of a core signature of MYC-responsive genes (Campbell and White, 2014; Ji et al, 2011). MYC is the archetypal member of its protein family, of which human MYCN and MYCL also belong. MYC is over-expressed and contributes to many different cancers as an oncogene. MYCN is most frequently overexpressed in cancers of neural origin and also regulates genes functioning in ribogenesis, whereas MYCL is most often overexpressed in small cell lung carcinomas (Boon et al, 2001; Duffy et al, 2021; Tansey, 2014). *Drosophila* has a single MYC gene. Like its vertebrate orthologues,

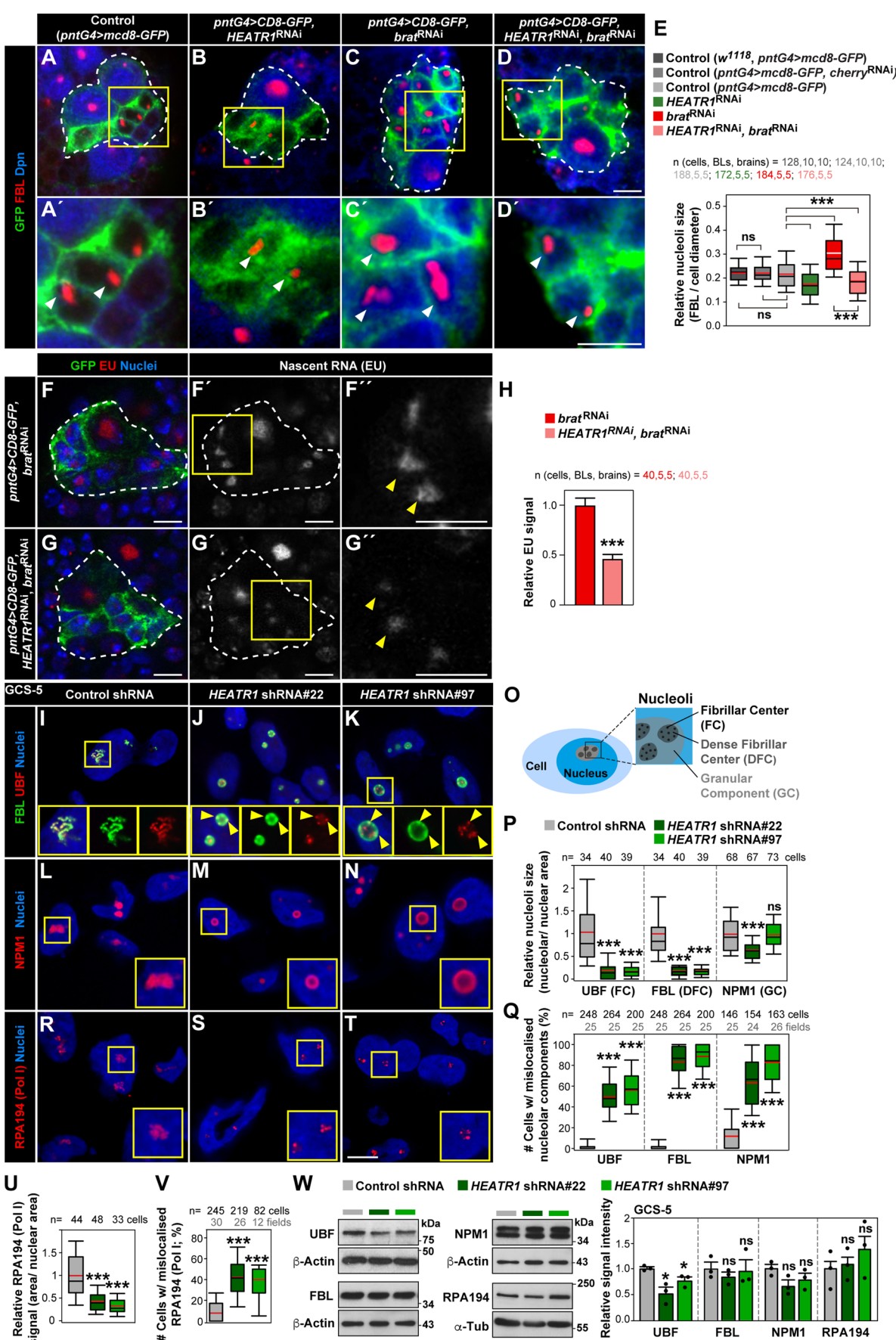

◀  **Figure 4. HEATR1 promotes nucleolar functional domains in *brat*-deficient brain TIC and GSCs.**

(A–E) Immunostainings of GFP, FBL and Dpn in type II NSC lineages expressing *CD8-GFP* (control) and *HEATR1*[RNAi], *brat*[RNAi] (tumour) or both *HEATR1*[RNAi], *brat*[RNAi] (*HEATR1*-deficient tumour) at 24 h ALH (A–D). Insets: higher magnification in (A'–D'). Arrowheads: nucleoli. Scale bars: 5 μm. Nucleoli size quantifications including in controls in which type II NSC lineages express *CD8-GFP* in the GD RNAi library host isogenic background or simultaneous *cherry*[RNAi] expression (see Appendix Fig. S3A–B'; Mann–Whitney tests; E; biological replicates: 5–10). (F–H) EU labelling (10 mM, 1 h) and immunostaining with GFP of type II NSC lineages expressing *CD8-GFP* and *brat*[RNAi] (tumour) or both *HEATR1*[RNAi], *brat*[RNAi] (*HEATR1*-deficient tumour) at 24 h ALH (F, G). Nuclei (DAPI). EU channel (monochrome; F',G'). Insets: higher magnification (F", G"; arrowheads: EU signal). Scale bars: 5 μm. EU signal quantification (H; Error bars: s.e.m.; Mann–Whitney test; biological replicates: 5). (I–V) Immunostainings of FBL and UBF (I–K), NPM1 (L–N) and RPA194 (Pol I; R–T) in GSCs 168 hpi with *HEATR1*-shRNAs (J, K, M, N, S, T) or control shRNAs (I, L, R). Nuclei (DAPI). Schematics: Nucleoli functional domains (O). Nucleoli size quantification provided by nucleoli domain (maximum) areas (P): 34–40 (UBF), 34–40 (FBL) and 67–73 (NPM1) cells from 3 biological replicates. Number of cells (%) with mislocalised nucleolar components (Q): 25 (UBF), 25 (FBL) and 24–26 (NPM1) cell images (fields) from 3 biological replicates. Quantification of Pol I (maximum) areas (U): 33–48 cells from 3 biological replicates. Cell numbers (%) with mislocalised Pol I signal (V): 12–30 fields from 3 biological replicates. Mann–Whitney tests. Scale bars: 10 μm; 5 μm in insets (arrowheads: mislocalised nucleolar components). (W) Immunoblots of GSCs 168 hpi with *HEATR1*-shRNAs or control shRNAs with indicated antibodies. β-Actin, α-Tubulin: loading controls. Quantifications of relative signals. Error bars: s.e.m. Biological replicates: 3. Unpaired two-tailed t-tests. Data information: Dashed lines: type II NSC lineages (A–D; F–G'). Box plots represent 25th and 75th percentiles, central black bands indicate medians, central red or white bands specify means, whiskers indicate 10th and 90th percentiles. ***$p \leq 0.001$; *$p \leq 0.05$; $p > 0.05$, ns (non-significant). Source data are available online for this figure.

it encodes a transcription factor that activates many targets, including genes involved in ribogenesis and controlling rRNA synthesis (Gallant, 2013; Grewal et al, 2005). MYC is overexpressed in *brat* NSC lineages, and its inhibition reduces *brat* brain tumour growth (Betschinger et al, 2006; Song and Lu, 2011; Zaytseva et al, 2020). We show that inhibiting *MYC* prevents *brat*-deficient TIC enlargement while not affecting divisions at 24 h ALH. By 48 h ALH, *MYC* inhibition prevents TIC overproliferation with mitotic numbers remaining at control levels (Fig. 6A–I, compare to Fig. 2P; see also Appendix Fig. S7A,B for mitotic index quantifications). *MYC* overexpression increases nucleolar size in NSC lineage cells (Song and Lu, 2011). In *brat*-depleted TICs, we show that inhibiting *MYC* reduces rRNA synthesis, indicating MYC's role in their ribogenesis (Fig. 6J–L). Since the effects observed upon *MYC* inhibition in *brat* TICs parallel those following *HEATR1* depletion, we sought to investigate if HEATR1's action involves MYC. MYC is expressed in NSCs and post-transcriptionally repressed by Brat in INP progeny (Dpn⁻) (Fig. 6M,M') but overexpressed in *brat*-deficient TICs (Betschinger et al, 2006; Zaytseva et al, 2020) already at 24 h ALH (Fig. 6O,O'). In controls, MYC remains repressed in INP progeny following *HEATR1* depletion (Fig. 6N,N'). In *brat*-deficient TICs, where MYC is overexpressed, *HEATR1* inhibition has no significant effect in its total levels (Fig. 6P–S), yet nucleolar MYC detected is reduced, as demonstrated by a small but significantly decreased signal in nucleoli marked by FBL antibodies (Fig. 6Q–T).

We next analysed HEATR1 and MYC in GBM cells. We focused on human MYC (c-Myc) since compared to MYCN it shares the highest orthology conservation with the *Drosophila* counterpart (Hu et al, 2011). GBM cells depleted of *HEATR1* show reduced MYC localisation in nucleoli (Fig. 7A–D) and no change in total MYC levels (Fig. 7E). MYC is predominantly found in nuclei, shuttling to and from the cytoplasm, and is subject to rapid turnover by the proteasome machinery. Upon MYC overexpression and proteasome inhibition, MYC can strongly localise to intranucleolar regions intermingling but in distinct sites from DFCs labelled by FBL, and this approach has been used to examine its nucleolar function in different cell types (Arabi et al, 2003; Arabi et al, 2005; Li and Hann, 2013; Murai et al, 2018). We observe MYC localising within nucleoli in 20% of GBM cells overexpressing MYC after proteasome inhibitor treatment (Fig. EV5A,J). Most other cells (42%) express MYC throughout the nuclei (Fig. EV5B,L) and the remaining show MYC in aggregates known as aggresomes (Fig.

EV5C,K), which are thought to be non-functional and arising due to the overwhelmed cell protein degradation system (Arabi et al, 2003; Li and Hann, 2013). Upon *HEATR1* knockdown, the proportion of cells with nuclear MYC or with MYC in agressomes is unchanged. Yet, a reduction in cells showing nucleolar MYC is seen with a concomitant increase in cells displaying MYC surrounding FBL accumulations but unable to enter nucleolar sites (Fig. EV5D–M).

MYC controls ribogenesis in both mammals and *Drosophila* by influencing expression of genes involved in rRNA generation and assembly into ribosomes. Yet, while no direct binding of MYC to *Drosophila* rDNA genes has been detected, mammalian MYC can bind directly to rDNA to activate Pol 1 mediated transcription (Arabi et al, 2005; Grandori et al, 2005; Grewal et al, 2005). To investigate if HEATR1 is involved in MYC's nucleolar function, we examined *47S* pre-rRNA levels. We were unable to overexpress HEATR1 in U87MG GBM cells, an issue reported in other cell lines (Fang et al, 2020) but could do it in both HeLA and 293T cells (Figs. 7F and EV5N, right panels). In addition, while we could not upregulate *47S* pre-rRNA upon MYC overexpression in U87MG cells, overexpressing MYC in either HeLA or 293T cell lines results in increased *47S* pre-rRNA levels, as expected (Figs. 7F and EV5N, left panels). We found that *HEATR1* overexpression alone is not sufficient to elicit changes in *47S* pre-rRNA levels, and is also unable to enhance increased *47S* levels upon MYC overexpression (Figs. 7F and EV5, left panels). We next sought to test in HeLa cells if the rise in *47S* pre-rRNA levels upon MYC overexpression could be altered by *HEATR1* inhibition. Indeed, HEATR1 depletion abolished the observed increase in *47S* pre-rRNA* (Fig. 7G). In all conditions MYC is highly overexpressed although the attained MYC overexpression levels were higher in controls than in cells with concomitant *HEATR1* inhibition. Yet, following HEATR1 depletion, *47S* pre-rRNA levels are dramatically reduced and back to those seen in cells with no MYC overexpression (Fig. 7G left and middle panels). Interestingly, while HEATR1 is required for rRNA transcription (Figs. 4 and 5), which depends on MYC function, it appears dispensable for the transcription of two known MYC targets, *RRN3* and *Pol1B*, not transcribed by Pol I (Campbell and White, 2014; Poortinga et al, 2011) (Fig. EV5O), suggesting it does not affect all MYC targets.

*HEATR1* was identified as a MYC target in other cell types (Furrer et al, 2010; Hulf et al, 2005; Seitz et al, 2011). In line with these studies, we observe that MYC overexpression in HeLa cells

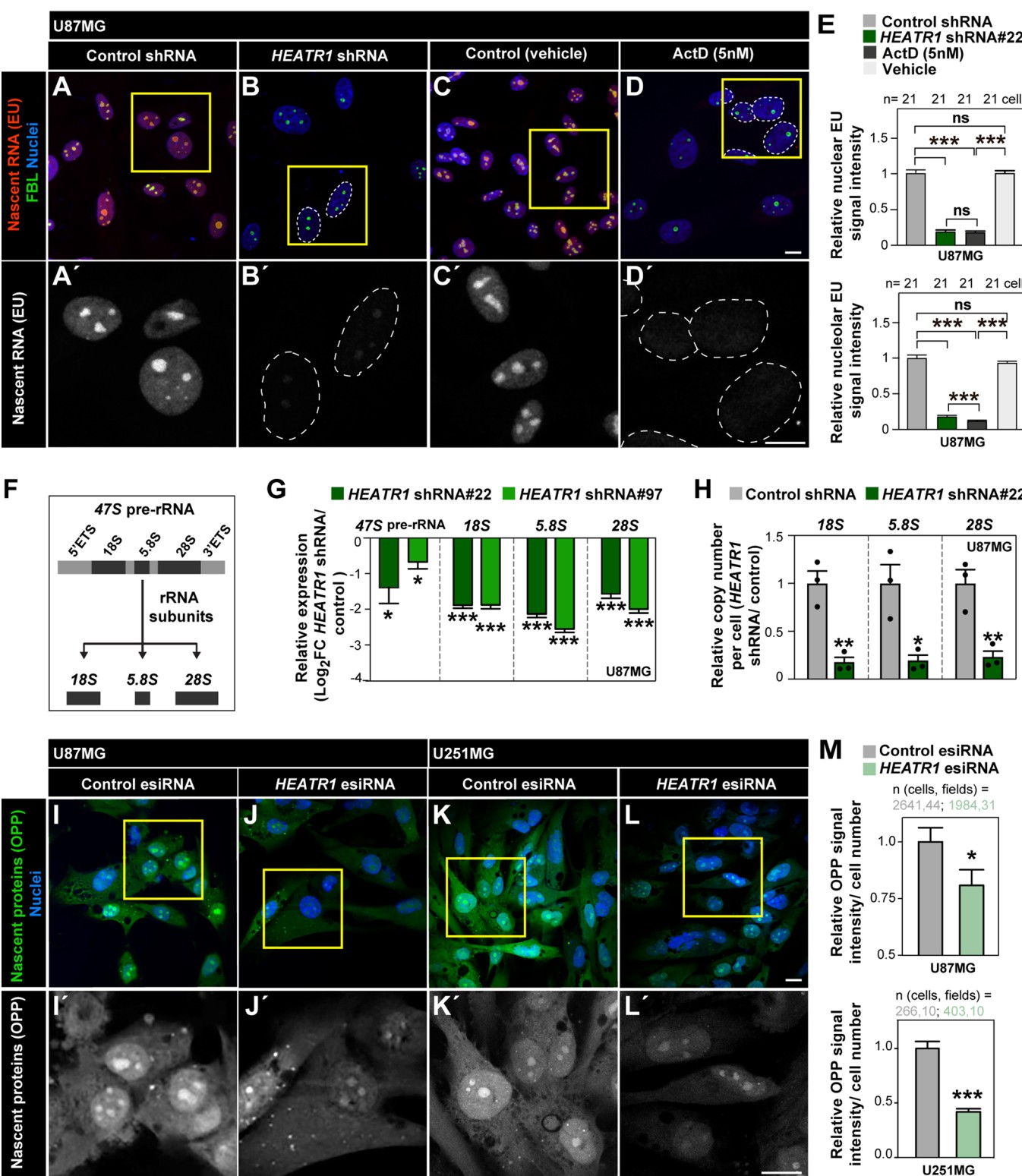

results in a small but significant upregulation of *HEATR1* (Fig. 7G, right panel). Similarly, in GBM cells, MYC overexpression also leads to an increase in *HEATR1* levels, suggesting HEATR1 may also be a MYC target in this context (Fig. 7H). On the other hand, HEATR1 harbours HEAT repeats that facilitate protein–protein interactions (Yoshimura and Hirano, 2016) and was detected as a putative MYC interactor in high-throughput proteomic approaches (Ewing et al, 2007; Heidelberger et al, 2018). We thus sought to test their possible physical interaction. Via co-immunoprecipitation (co-IP) assays using 293T cells expressing HA-tagged MYC and

**Figure 5. *HEATR1* inhibition reduces ribogenesis and protein synthesis in GBM cells.**

(A–E) EU-labelling (1 mM, 1 h) and immunostaining with FBL in U87MG cells 168 hpi with *HEATR1*-shRNA (B) or control shRNA (A) or treated with ActD (D) or vehicle (C) for 168 h. Nuclei (DAPI). Insets: higher magnifications in (A'–D') (monochrome). Dashed lines: nuclei (B', D'). Scale bars: 10 µm. Nuclear (upper panel) and nucleolar (lower panel) EU signal quantification (E; 21 cells per condition from 3 biological replicates). Error bars: s.e.m. Unpaired two-tailed t-tests. (F) Schematics: *47S* pre-rRNA and resulting *18S, 5.8S* and *28S* rRNA subunits. (G, H) RT-qPCR analysis of *47S* pre-RNA, *18S, 5.8S* and *28S* rRNAs in U87MG cells 168 hpi with *HEATR1*-shRNAs versus control shRNAs. Fold change (FC) expression levels (G; biological replicates: 3, technical replicates: 2) and copy number per cell quantifications (H; biological replicates: 3). Error bars: s.e.m. Unpaired two-tailed t-tests. (I–M) OPP-labelling (20 µM, 30 min) in GBM cell lines (U87MG, U251MG) 48 hpt with *HEATR1*-esiRNA (J, L) or control *GFP*-siRNA (I, K). Nuclei (DAPI). Insets: higher magnifications in (I'–L') (monochrome). OPP signal quantification in U87MG (M, upper panel): 31–44 cell images (fields) from 3 biological replicates. OPP signal quantification in U251MG (M, lower panel; 10 fields per condition from 3 biological replicates). Error bars: s.e.m. Mann–Whitney tests. Data information: ***p ≤ 0.001; **p ≤ 0.01; *p ≤ 0.05; p > 0.05, ns (non-significant). Source data are available online for this figure.

Flag-tagged HEATR1, we demonstrate that HA-MYC co-IPs Flag-HEATR1 (Fig. 7I left panel) and conversely Flag-HEATR1 co-IPs HA-MYC (Fig. 7I right panel), with no association found with respective HA or Flag control proteins. The results show that HEATR1 physically binds MYC.

Collectively, our data suggest that during early tumourigenesis brain TICs undergo dramatic reprogramming in which ribosome biogenesis changes play a central role. The findings lead us to propose a model whereby upon neoplastic transformation, brain TICs upregulate HEATR1, which in turn binds MYC and promotes its localisation and action in nucleoli sites. High HEATR1 levels are required for an increase in ribogenesis promoting TIC development and transition into tumour growth (Fig. 7J).

## Discussion

Massive efforts in recent years increased our knowledge of TIC properties, tumour development, heterogeneity and environment interaction (Ayob and Ramasamy, 2018; Puisieux et al, 2018). Most profiling and functional screen approaches rely on tissue dissociation, cell sorting and cultures (Lawson et al, 2018; Macklin et al, 2020; MacLeod et al, 2019). We expose a transcriptome profiling of TICs obtained directly from live brains. The analysis of traceable individual cells taken directly from living tissues at specific timepoints allows to precisely probe the transcriptional control of critical cell fate changes (Barros and Bossing, 2021; Bossing et al, 2012; Gil-Ranedo et al, 2019). We focused on TICs transitioning into tumour growth using the *Drosophila brat* NSC-derived tumour model (Bello et al, 2006; Betschinger et al, 2006; Bowman et al, 2008; Caussinus and Gonzalez, 2005; Landskron et al, 2018). Our transcriptome data show high conservation with mammalian transcripts and is enriched in genes for which the orthologues were identified in a genome-wide functional screen as required for GSC growth (MacLeod et al, 2019), suggesting it is a valuable resource in tumourigenesis research.

Many processes are linked to tumour growth including inactivation of cell cycle regulators, oncoprotein overexpression, genomic instability, epigenetic modifications, oxidative DNA damage and apoptosis evasion (Dewhurst, 2020; Feitelson et al, 2015; Fridman and Tainsky, 2008). Our *brat* TIC profiling reveals that metabolic and proteostasis alterations are core to cells reprogramming into initial stages of tumour growth. Changes in such processes are known as adaptations of formed tumours to meet growth demand and environment changes, but how these contribute to early tumour development is only emerging (Bastola et al, 2018; Cairns et al, 2011). We detected OXPHOS prominently among metabolic pathways. OXPHOS is involved in the maintenance of several human cancers (Janiszewska

et al, 2012; Rao et al, 2019) and a bioenergetic switch to OXPHOS was recently proposed to be a primary feature upon *brat* brain TIC commitment to tumourigenesis (Bonnay et al, 2020; van den Ameele and Brand, 2019). Within genes we identified in the proteostasis group, those related to ribosomes and their biogenesis are most abundant. Several inherited mutations affecting ribogenesis are associated with higher cancer risk, and alterations in rRNA synthesis players are linked to cancer stem cells (Mannoor et al, 2014; Orsolic et al, 2020; Pelletier et al, 2018). Yet, to our knowledge, our findings provide the first strong evidence that a ribogenesis boost precedes and is required for the transition of brain TICs into tumour growth. First, we show that hypertrophy of nucleoli of *brat*-deficient brain tumours is already visible in TICs during early stages of development prior to overproliferation onset. Enlarged nucleoli is recognised in cancer pathology to correlate with poor prognosis and reflects enhanced ribogenesis (Derenzini et al, 2009). We then demonstrate that *HEATR1* depletion is sufficient to prevent nucleolar enlargement and the elevated nascent rRNA levels in brain TICs at this early stage, as well as inhibiting cell size increase. In contrast, in normal type II NSC lineages, *HEATR1* inhibition has only a marginal effect on nucleoli size and does not affect cell size nor proliferation rate at early stages. By the time *brat*-deficient TICs start to overproliferate, *HEATR1* inhibition continues to have no effect in the proliferation of non-tumour lineages but it delays the overgrowth onset of *brat* tumours, with TIC proliferation rates remaining at levels comparable to those of controls. At late stages, *HEATR1* depletion results in dramatically reduced tumours, however, it also decreases proliferation of control NSC lineages highlighting its requirement for normal brain development. This is consistent with a report identifying *HEATR1*, as well as *MYC* (see discussion below), as the highest scoring genes among a group of 68 ribogenesis-associated factors screened for ability to promote rRNA synthesis in a normal (non-tumour) breast cell line (Bryant et al, 2022). Together with the facts that *HEATR1* is overexpressed in *brat* brain TICs and that *HEATR1* depletion has significantly more pronounced effect in TICs' nucleoli even prior to tumour growth onset, our data indicate that *brat*-deficient TICs may have a higher dependence on HEATR1's function in ribogenesis than their normal non-tumour cell counterparts. HEATR1 is also overexpressed in patient-derived GSCs compared to normal brain tissue or NSCs, and its expression inversely correlates with glioma patient survival. Moreover, HEATR1 is required for GSC nucleoli size, organisation and tumourigenic capacity, as well as for GBM cell rRNA and protein synthesis, suggesting it may promote GBM tumourigenesis via boosting ribogenesis.

HEATR1 has been proposed to be required for ribogenesis in osteosarcoma U2OS cells and during zebrafish CNS development. Its depletion in U2OS cells leads to p53 accumulation and p53-dependent cell cycle arrest (Turi et al, 2018), and in zebrafish it

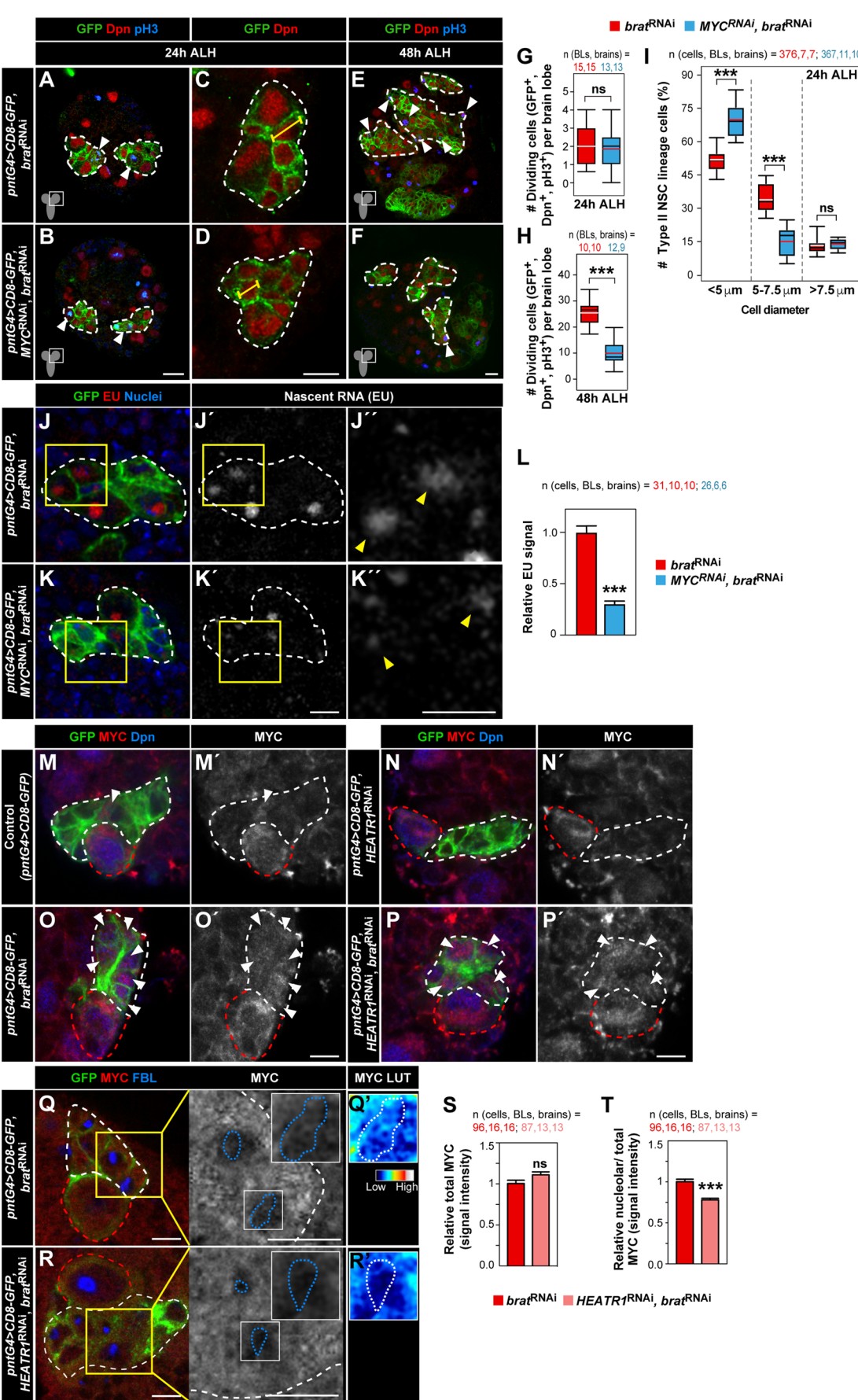

**Figure 6. Nucleolar MYC in *brat*-deficient brain TICs requires HEATR1.**

(A–I) Immunostainings of GFP, Dpn and pH3 (A–F; GFP and Dpn channels only in C and D) in type II NSC lineages expressing *CD8-GFP* and *brat*[RNAi] (tumour) or *MYC*[RNAi], *brat*[RNAi] (*MYC*-deficient tumour) at 24 h (A–D) and 48 h (E, F) ALH. Yellow lines: maximum cell diameter. Scale bars: 10 μm. Quantification of mitotic cells (GFP⁺Dpn⁺pH3⁺) per brain lobe at 24 h (G; biological replicates: 13–15) and 48 h ALH (H; biological replicates: 10–12). See mitotic cell index quantifications (Appendix Fig. S7A, B). Quantification of cell sizes in type II NSC lineages per brain lobe at 24 h ALH (Unpaired two-tailed t-tests; I; biological replicates: 7–11). Arrowheads: pH3⁺ cells. Box plots represent 25th and 75th percentiles, central black bands indicate medians, central red or white bands specify means, whiskers indicate 10th and 90th percentiles. (J–L) EU labelling (10 mM, 1 h) and immunostaining with GFP of type II NSC lineages expressing *CD8-GFP* and *brat*[RNAi] (tumour) or *MYC*[RNAi], *brat*[RNAi] (*MYC*-deficient tumour) at 24 h ALH. Nuclei (DAPI). EU channel (monochrome; J′–K″). Insets: higher magnification (J″, K″). Scale bars: 5 μm. EU signal quantification (L; Error bars: s.e.m.; Mann–Whitney test; 26–31 cells from 6–10 biological replicates). Yellow arrowheads: EU signal. (M–P) Immunostainings of GFP, MYC and Dpn in type II NSC lineages expressing *CD8-GFP* (control) and *HEATR1*[RNAi], *brat*[RNAi] (tumour) or both *HEATR1*[RNAi], *brat*[RNAi] (*HEATR1*-deficient tumour) at 24 h ALH. Arrowheads: Dpn⁺Myc⁺ INP progeny. Scale bars: 5 μm. (Q–T) Immunostainings of GFP, MYC and FBL in type II NSC lineages of *brat*[RNAi] (tumour) and *HEATR1*-deficient tumour at 24 h ALH. Scale bars: 5 μm. Insets: higher magnification for MYC channel (monochrome). Quantification of total (Unpaired two-tailed t-test; S; 87–96 cells from 13–16 biological replicates) and nucleolar versus total (Unpaired two-tailed t-test; T; 87–96 cells from 13–16 biological replicates) MYC signal. Error bars: s.e.m. Q′ and R′ showing higher magnification and with pseudocolour linear Lookup Table (LUT) for enhanced visualisation. Dotted lines: nucleoli (Q, R in blue; Q′, R′ in white). Data information: Dashed lines: type II NSC lineages (NSCs, red in M–R; NSC progeny, white in M–R). ***p ≤ 0.001; p > 0.05, ns (non-significant). Source data are available online for this figure.

triggers p53-dependent apoptosis (Azuma et al, 2006). In non-small cell lung cancer cells, *HEATR1* depletion also induces p53-dependent apoptosis (He et al, 2019) and in gastric cancer lines it reduces proliferation and survival with increased phosphorylated p53, p38 MAPK, Chk2 and IKBa expression, suggesting it may function via interaction with different pathways (Zhao et al, 2020). We observed no effect on survival of type II NSC lineages, *brat*-deficient tumours or GBM cells following *HEATR1* inhibition, and only a small rise in apoptosis is seen in one of the GSC lines analysed. In addition, no significant changes in p53 levels are detected despite a marked reduction in rRNA generation. While we cannot exclude involvement of p53 signalling independent of its accumulation in cells, our data and previous reports indicate that HEATR1's functions may be cell context-dependent. In support, HEATR1 expression in glioma negatively correlates with patient survival but a positive correlation was also found in pancreatic ductal adenocarcinoma where it is thought to aid sensitization to chemotherapeutic gemcitabine via AKT signalling inactivation (Fang et al, 2020; Liu et al, 2016). Our findings indicate that HEATR1 may function in ribogenesis of brain TICs at least in part via binding and promoting MYC nucleolar localisation.

MYC is a powerful oncoprotein and master regulator of ribogenesis and growth (Duffy et al, 2021). Brat in *Drosophila* brains, and TRIM3 in GBM cells, supress *MYC* expression, and *MYC* inhibition reduces both *brat* tumour growth and GSC tumourigenic potential in vitro and when xenotransplanted in mice (Betschinger et al, 2006; Chen et al, 2014; Wang et al, 2008). We detected ectopic MYC in *brat*-deficient TICs even prior to tumour growth and show that *MYC* depletion during this early stage of TIC development mirrors that of *HEART1*, precluding enhanced rRNA synthesis, cell enlargement and delaying the start of tumour growth. Interestingly, in a lymphomagenesis mouse model entailing *MYC* constitutive expression, an increase in cell size is observed in pre-transformed B cells (Iritani and Eisenman, 1999), suggesting that MYC-dependent cell enlargement is a process that may be conserved in the development of some mammalian cancers.

*HEATR1* contains a canonical Myc-binding E-box (CACGTG) in its promoter and has been proposed as a MYC transcriptional target in a genome-wide analysis of MYC-binding sites in Burkitt lymphoma cell lines by chromatin immunoprecipitation followed by sequencing (ChIP-Seq) (Seitz et al, 2011) and in *Drosophila* S2 cells (Furrer et al, 2010; Hulf et al, 2005), with these studies also reporting that *MYC* inhibition decreases *HEATR1* expression. We

detect a small but significant increase in *HEATR1* expression upon MYC overexpression in GBM cells. It is thus interesting to postulate that *HEATR1* may also be a MYC target during brain tumour development. On the other hand, we demonstrate that HEATR1 protein can physically bind MYC and show that HEATR1 is required for MYC nucleolar localisation, suggesting how it may contribute to MYC's ribogenesis control. Our data thus point to an interplay between MYC and HEATR1 proteins, in addition to a potential genetic interaction previously reported.

In both *Drosophila* and mammals, MYC stimulates rRNA synthesis and ribosome assembly. Yet, unlike in mammals, *Drosophila* rDNA loci lack the consensus E-box MYC binding sites and MYC has been reported not to bind rDNA directly (Grewal et al, 2005; Orian et al, 2005). Thus, while MYC regulation of ribogenesis seems indirect in *Drosophila*, in mammals it is both indirect and direct via rDNA binding (Arabi et al, 2005; Grandori et al, 2005; Grewal et al, 2005). The histone chaperon NPM1 has also been shown to bind MYC and be required for its nucleolar localisation and activation of rDNA transcription in *p53* mutant mouse embryonic fibroblasts (Li and Hann, 2013). Of note, while our work was in revision, HEATR1 was published to be upregulated by the Target of Rapamycin Complex 1 signalling and promote hepatocellular carcinoma (HCC) development (Yang et al, 2023). HEATR1 was shown to bind rRNA and its inhibition to disrupt ribogenesis and growth of HCC cells while limited effects were observed in proliferation of immortalized normal hepatic cells. The study underlines and supports the main result of our research on the pivotal role of HEATR1 in rRNA transcription and promotion of oncogenesis. Yet, the mechanisms proposed diverge although may not be mutually exclusive. The divergence may reflect the difference in tumours and models. Yang et al propose that *HEATR1* inhibition increases nuclear proteasome activity resulting in reduced nucleolar NPM1 and subsequently preventing Myc nuclear localisation (Yang et al, 2023). In GBM cells, despite nucleolar stress induced by *HEATR1* depletion, we detect no significant changes in NPM1 protein levels but observe a marked reduction in the ratio of nucleolar to nuclear MYC localisation. In summary, HEATR1 is crucial for ribogenesis and has been increasingly implicated in the development of different cancers but its modes of action may vary not only between tumour types but also between transition into tumourigenesis and progression of established cancers. HEATR1's binding and modulation of MYC nucleolar localisation and its requirement for rRNA generation lead us to

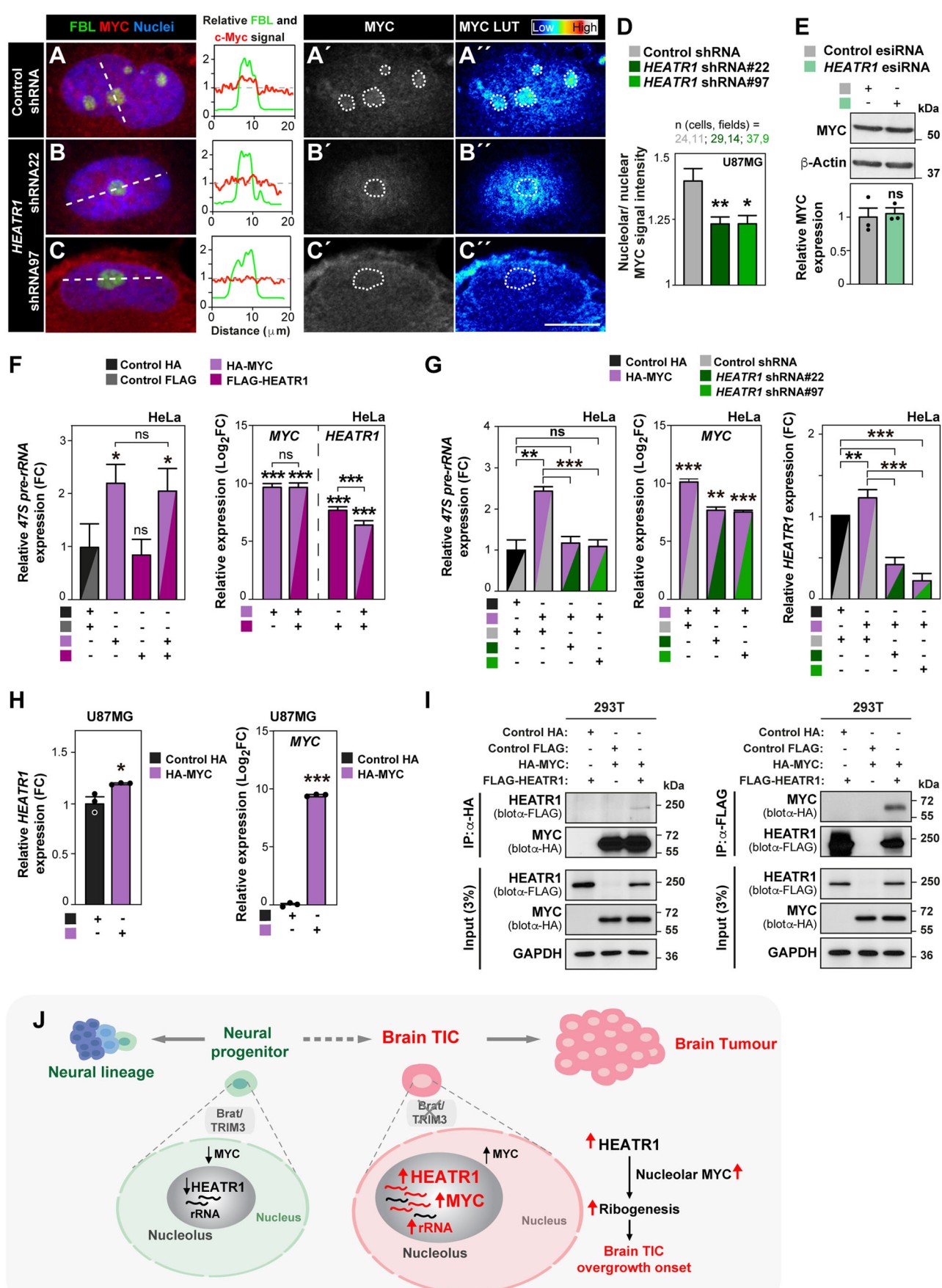

Figure 7.  HEATR1 promotes MYC nucleolar localisation and ribogenesis control in GBM cells.

**(A–D)** Immunostainings of FBL and MYC in U87MG GBM cells 168 hpi with *HEATR1*-shRNAs (**B**, **C**) or control shRNAs (**A**). Nuclei (DAPI). Corresponding relative signal intensity profiles over nucleus and nucleoli (dashed lines). MYC channel (monochrome, **A'–C'**; pseudocolour linear Lookup Table applied, **A"–C"**). White dotted lines: nucleoli. Scale bars: 10 µm. Nucleolar versus nuclear MYC signal quantifications: D, unpaired two-tailed t-tests, 9–14 cell images (fields) from 3 biological replicates. **(E)** Immunoblots of MYC in U87MG cells 48 hpt with *HEATR1*-esiRNA or control *GFP*-esiRNA. β-Actin: loading control. Relative MYC signal quantification. Biological replicates: 3. Unpaired two-tailed t-tests. **(F)** RT-qPCR analysis of *47S* pre-RNA in HeLa cells transfected with HA-tagged MYC (HA-MYC) or control HA vector, and Flag-tagged HEATR1 (Flag-HEATR1) or control Flag vector (left panel). RT-qPCR analysis of *MYC* and *HEATR1* in HeLa cells expressing Flag-HEATR1 or HA-MYC versus controls in same assays (right panel). Fold change (FC). Biological replicates: 3; technical replicates: 2. Unpaired two-tailed t-tests. **(G)** RT-qPCR analysis of *47S* pre-rRNA in HeLa cells 168 hpi with *HEATR1*-shRNAs or control shRNAs and transfected with HA-tagged MYC (HA-MYC) or control vector (left panel). RT-qPCR analysis of *MYC* (middle panel) and *HEATR1* (right panel) in same assays. Fold change (FC). Biological replicates: 3; technical replicates: 2. Unpaired two-tailed t-tests except Mann–Whitney tests in *47S* analysis on double control shRNA, control HA expressing samples and in *MYC* analysis on double *HEATR1* shRNA#22, HA-MYC expressing samples. **(H)** RT-qPCR analysis of *HEATR1* in GBM U87MG cells transfected with HA-tagged MYC (HA-MYC) or control vector (left panel). RT-qPCR analysis of *MYC* upon HA-tagged MYC (HA-MYC) expression relative to control in the same assays is also shown (right panel). Fold change (FC). Biological replicates: 3. Unpaired two-tailed t-tests. **(I)** Co-IP assays of HEATR1 with MYC in 293T cells expressing HA-MYC, Flag-HEATR1, control Flag or HA proteins as indicated. Lysates and HA-purified (top panel) or Flag-purified (bottom panel) immunoprecipitates analysed by western-blot with indicated antibodies. GAPDH: loading control (n = 4 biological replicates: n = 2 HA- and n = 2 Flag-purified immunoprecipitates). **(J)** Proposed model of HEATR1 and MYC action in ribogenesis required for brain TIC transition into tumour growth. Data information: Error bars: s.e.m.; ***p ≤ 0.001; **p ≤ 0.01; *p ≤ 0.05; p > 0.05, ns (non-significant). Source data are available online for this figure.

propose it as a molecular mechanism underlying ribogenesis reprogramming of brain TICs, shedding new light on the transition into brain tumour growth (Fig. 7J). Given that ribosome biogenesis is one of the most energy-consuming cellular processes (Pelletier et al, 2018), the specific molecular signals to meet the energy demanded by the dramatic changes in ribogenesis during early TIC development is open to future research.

# Methods

## *Drosophila* strains and genetics

*brat^{K06028}* (Arama et al, 2000) was obtained from the Kyoto *Drosophila* Genetic Resource Consortium (114346) and rebalanced over *CyO*, P(GAL4-*twi.G*)2.2. *UAS-l(2)k09022-RNAi* (*UAS-HEATR1-RNA*; 17000) (Neumuller et al, 2011), *UAS-MYC-RNAi* (2947) (Rust et al, 2018) and *w^{1118}* (60000; isogenic host strain for the GD RNAi library that includes lines 17000 and 2947) were obtained from the Vienna *Drosophila* Resource Center (VDRC). *UAS-brat-RNAi* (34646) (Reichardt et al, 2018) and *UAS-CD8-GFP* (5137) were obtained from the Bloomington *Drosophila* Stock Center (BDSC). *UAS-Cherry-RNAi* (BDSC stock 35785) expresses dsRNA for RNAi targeting mCherry in the VALIUM20 vector, the same vector used to generate line 34646. *pointed-gal4* (*pnt-Gal4*) was a kind gift from Y. Jan (Zhu et al, 2011). For single-cell harvesting, *UAS-CD8-GFP* was recombined with *brat^{K06028}* and combined with *pnt-Gal4*. Type II NSC lineages-targeted RNAi assays made use of *pnt-Gal4* combined with *UAS-CD8-GFP*. For double *brat*- and *HEATR1*-deficient assays, combined *UAS-brat-RNAi* and *UAS-l(2)k09022-RNAi* transgenics were generated. For double *brat*- and *MYC*-deficient assays, *UAS-brat-RNAi* and with *UAS-MYC-RNAi* transgenics were combined. *Drosophila* husbandry: *Drosophila* lines were kept in standard fly food at 25 °C. Egg collections and larvae rearing were performed on agar juice plates (21 g agar, 200 ml of grape juice per l of water) supplemented with yeast paste. Egg lays were collected in 1 h time-windows.

## Human tissue

26 GBM, 9 grade II diffuse astrocytoma and 19 non-tumour anonymised brain tissue samples were obtained from University

Hospitals Plymouth NHS Trust as part of BRAIN UK (License 14/004) (Nicoll et al, 2022). *IDH* status of tumour samples determined by immunostaining against the common R132H IDH1 mutant protein. See Appendix Table S1 for sample details.

## Cell cultures and drug treatment

U87MG (Sigma, 89081402), U251MG (Sigma, 09063001) and 293T (ATCC, CRL-3216) human cell lines were maintained in complete standard MEM with 10% FBS (One shot, Gibco). GSC lines 5 and 8 were kind gifts from M. Izquierdo and J.M. Almendral and cultured as non-adherent tumourspheres as described (Gil-Ranedo et al, 2021; Gil-Ranedo et al, 2011). GSC *IDH* status was confirmed with immunostaining against R132H IDH1 mutant protein. Briefly, GSCs were maintained in complete DMEM/F12 (1:1, Gibco) with glutamax (1x, Gibco), N-2 (1x, Gibco), 2 µg/ml heparin (Acros Organic), 20 ng/ml EGF (Peprotech), 20 ng/ml FGF-basic (Peprotech), non-essential amino acids (1x, Gibco), and Penicillin-Streptomycin (Pen-Strep; Gibco). 6-week human foetal forebrain-derived NSC line (Glioma Cellular Genetics Resource, University of Edinburgh, FT3528 FB P5) was cultured in DMEM/F12 (1:1, Gibco) with glucose (0.14%, Gibco), non-essential amino acids (1x, Gibco), Pen-Strep, BSA (0.012%, Gibco), 2-Mercaptoethanol (0.1 mM, Gibco), N-2 (1x, Gibco), B27 (1x, Gibco), 10 ng/ml mouse EGF (Peprotech), 10 ng/ml human FGF-basic (Peprotech) and laminin (1 µg/ml, Sigma). Cultures were used between passage 10 and 20, as recommended by the European Collection of Authenticated Cell Cultures, and grown at 37 °C, 5% CO$_2$. Cell lines were not recently authenticated as they were purchased from commercial sources providing pathogen-free and identity-certified cell lines or were derived previously by JGR as indicated and were regularly monitored for mycoplasma contamination via DAPI staining (Russell et al, 1975; Young et al, 2010). In indicated assays, cells were treated with 5 µM MG132 (Sigma, 47491) or 0.05% DMSO (vehicle; Corning) for 3 h; or were treated with 5 nM Actinomycin D (Abcam, ab141058) or 0.0002% DMSO (vehicle; Corning) for the time noted. When indicated in GSC soft agar assays, 5 nM Actinomycin D or vehicle (0.0002% DMSO) were added to the 2 ml of complete DMEM/F12 medium with 0.35% agar and incubated for 10 days. After 5 of the 10 days, 0.8 mL of fresh DMEM/F12 medium containing 5 nM Actinomycin D or vehicle were added to mitigate media evaporation.

## Single-cell transcriptomics and bioinformatics

Single INP cells adjacent to the larger type II NSCs were individually harvested from live, freshly dissected 24 h ALH brains expressing *CD8-GFP* specifically in type II NSC lineages under control (*UAS-CD8-GFP; pnt-Gal4*) or *brat* (*UAS-CD8-GFP, brat06028; pnt-Gal4*) backgrounds. Single type II NSCs from analogous control and *brat* brains were also harvested and their transcriptome used for data normalisation process only. Single INP and type II NSC harvest, mRNA isolation, cDNA generation and microarray hybridization and scanning were performed as described (Barros and Bossing, 2022; Bossing et al, 2012). Briefly, single cells were placed in individual tubes containing 0.3 μL anchored polyT (5′-AAGCAGTGGTATCAACGCAGAG TACT$_{(26)}$VN-3′, 10 μM) and 0.3 μL SM (5′-AAGCAGTGGTAT CAACGCAGAGTACGCrGrGrG-3′, 10 μM) primers, 0.4 μL RNase inhibitor (Superase, Ambion) and 2 μL lysis buffer (10% Nonidet P-40, 0.1 M DTT in DEPC-treated ultrapure water) and processed in less than 20 min. After centrifugation (14,000 rpm, 1 min, 4 °C) and annealing (3 min, 70 °C), samples were snap frozen. For reverse transcription, 1.5 μL of mix 1 (1 μL first strand buffer, Invitrogen, and 0.5 μL dNTPs, 10 mM) and 0.5 μL of mix 2 (3 μL Superscript II, Invitrogen, 0.5 μL Superase, Ambion) were added (90 min, 37 °C) followed by thermal inactivation (10 min, 65 °C), RNase H (Roche) treatment (20 min, 37 °C) and again thermal inactivation (15 min, 65 °C). Single cDNA samples obtained were next amplified via using 2 μL of nested primer (5′-AAGCAGTGGTATCAACGCA-GAGT-3′, 10 μM), 2 μL 10 mM dNTPs, 0.5 μL Long Expand polymerase (Roche), 5 μL buffer 1 and 34.5 μL of water. PCR program: 3 min 95 °C, 5 min 50 °C, 15 min 68 °C) followed by 24 cycles (20 s 95 °C, 1 min 60 °C, 7 min plus 10 sec per cycle 68 °C). Single-cell cDNAs showing clear banding patterns on agarose gels were subjected to PCR using *ase* primers (see Appendix Table S2). All samples were *ase* negative except one control INP deemed mature (mINP; Fig. 1A). cDNAs were sent for microarray analysis (FlyChip, University of Cambridge). 1 μg of each sample were Klenow-labelled using BioPrime DNA Labelling System (Invitrogen) in the presence of Cy3- or Cy5-dCTP (GE Healthcare) for 2 h 37 °C. Unincorporated dye and nucleotides were removed using AutoSeq G-50 columns (GE Healthcare), following manufacturer instructions. Cy3- and Cy5-labelled pairs of samples (*brat* INP versus *brat* type II NSC, 3 pairs; control iINP versus control type II NSCs, 3 pairs; control mINP versus control type II NSC, 1 pair) were combined with salmon sperm DNA as blocking agent and co-hybridized (16 h, 51 °C) in a HybStation hybridization station (Digilab Genomic Solutions) on long oligonucleotides FL003 microarrays. Post-hybridization washes were performed according to Full Moon Biosystems protocols. Detailed protocols for labelling, hybridization and washing can be requested from the Cambridge Systems Biology Centre UK, University of Cambridge (https://www.sysbiol.cam.ac.uk). Arrays were scanned at 5 μm resolution (GenePix scanner, Axon Instruments) using optimised PMT gain settings for each channel.

PCA was performed using singular value decomposition with imputation method on normalized values of individual samples, via ClustVis web tool (Metsalu and Vilo, 2015).

DRSC Integrative Ortholog Prediction Tool (DIOPT) v8.5 (Hu et al, 2011) was accessed on 2023-03 for gene orthology analysis. KOBAS v3.0 web server (Bu et al, 2021) was accessed on 2023-03 for data enrichment analysis of Kyoto Encyclopedia of Genes and Genomes (KEGG) pathways applying a Fisher exact test and Benjamini and Hochberg FDR correction (Benjamini and Hochberg, 1995). For analysis of *Drosophila* transcriptome data, genes with FDR < 0.1 were used as input and FL003 microarray gene list as background. For analysis of corresponding human gene dataset, orthologues (DIOPT scores ≥ 5) were used as input and the human genome as background. A minimum term size cut off of 6 genes was applied.

*Drosophila* protein–protein interaction networks generated using Cytoscape v3.6.1 (Shannon et al, 2003) via stringApp v1.4.2 (Doncheva et al, 2019). Experimental-based data only used as source; minimum required interaction score of 0.4 applied.

*HEATR1* expression analysis in grade II and IV gliomas used TCGA_GBMLGG (Ceccarelli et al, 2016) database via GlioVis (Bowman et al, 2017). For *TRIM3* and *HEATR1* expression correlation in GBM, RNAseq expression values were compiled from TCGA_GBM dataset (Cancer Genome Atlas Research et al, 2013) via GlioVis (Bowman et al, 2017), and Ivy Glioblastoma Atlas Project (Ivy GAP) (Puchalski et al, 2018). Data was normalized to z-scores.

*TRIM3* copy number alterations study in low- and high-grade glioma conducted using unique patient TCGA_LGG, TCGA_GBM and TCGA_GBMLGG collated datasets. Deletions detected using GISTIC2.0 algorithm (Mermel et al, 2011) and data retrieved via GlioVis (Bowman et al, 2017).

For gene set enrichment analysis (GSEA) and visualization of results, the R packages fgsea (Korotkevich et al, 2021) and clusterProfiler (Wu et al, 2021) were applied. As ranked list, results of the genome-wide CRISPR-Cas9 GSC fitness screen (MacLeod et al, 2019) ordered by corresponding Bayes Factor were used after discarding *EGFP*, *luciferase* and *LacZ* internal controls. As gene set, the list of human orthologue genes (DIOPT score ≥ 5 from our *Drosophila* transcriptome dataset FDR < 0.1) was used. Empirical *p*-value for enrichment was estimated based on $10^7$ random permutations.

Survival analyses used TCGA LGG_GBM (Ceccarelli et al, 2016), Rembrandt (Madhavan et al, 2009) and CGGA (Zhao et al, 2021) datasets via GlioVis (Bowman et al, 2017). Samples with no tumour grade information were removed. Kaplan-Meier survival curves with logrank tests compared the 25% of samples with lower and higher *HEATR1* expression. *HEATR1* expression analysis in GBM CSC versus non-CSC clusters used Ivy GAP data, in which clusters were identified by high or low expression of 17 CSC reference probes by in situ hybridization, isolated by laser microdissection and subjected to RNAseq (Puchalski et al, 2018). *TRIM3* and *HEATR1* expression analysis in GBM regions (leading edge, infiltrating tumour, cellular tumour and pseudopalisading cells around necrosis) used RNAseq expression z-scores, Ivy GAP (Puchalski et al, 2018). Heatmap generated by clustering z-scores using Heatmapper web server (Babicki et al, 2016).

## cDNA generation and RT-qPCR

Single-cell cDNA generation is described above. For cDNA production from human tissue and cell lines, total RNA was extracted using TRI reagent (Invitrogen) or RNAqueous Micro kit (Invitrogen) following manufacturer's instructions. For reverse transcription, 2 μL of polyT primer (10 μM) and 2 μL of SM (10 μM) primer were added to 500 ng of RNA in 20 μL of DEPC-

treated water, annealed (3 min 65 °C) and snap-frozen. Each sample was next incubated (90 min 42 °C) with 1 μL of Supercript II (Invitrogen), 8 μL of 5x First Strand buffer, 4 μL of 0.1 M DTT, 2 μL 10 mM dNTPs and 1 μL of Superase (Invitrogen) followed by enzyme inactivation (15 min 65 °C). RNA was digested by adding 1 μL RNase H (Thermo Scientific) and 4 μL 10x RNase H buffer (20 min 37 °C) and reaction stopped via enzyme inactivation (15 min 65 °C). RT-qPCRs (5 ng cDNA per sample) used SYBR-Green and primers indicated in Appendix Table S2 on a StepOnePlus thermal cycler (Applied Biosystems). RT-qPCRs using *EN1* primers on GSCs yielded undetermined CT values preventing *EN1* expression analysis in these cells. For rRNA RT-qPCRs, cDNA was generated using High-Capacity cDNA Reverse Transcription kit including random hexamers (Applied Biosystems) following manufacturer's instructions and 50 ng cDNA generated used per reaction. *ribosomal protein 49* (*rp49*) and *Ribosomal protein 32* (*RPL32*) were employed as internal calibrators for reactions using *Drosophila* and human samples, respectively. To calculate *5.8S*, *18S* and *28S* rRNA copy numbers, RT-qPCR standard curves were generated in triplicate for each target by serial dilution of known amounts of ethanol-precipitated PCR products from U87MG cDNA and primers indicated in Appendix Table S2. Sample masses were inferred from standard curves, converted into DNA copy numbers as described (Ma et al, 2021) and normalized to copy number per cell.

## Immunohistochemistry, TUNEL and EU assays in tissue

*Drosophila* brain immunohistochemistry was performed as described (Gil-Ranedo et al, 2019). Briefly, larval brains were dissected in PBS and fixed for 20 min in 4% formaldehyde in PBS with 5 mM MgCl$_2$ and 0.5 mM EGTA (early stages) or 10 mM MgCl$_2$ and 1 mM EGTA (late stages), followed by washes in PBS (2 × 10 min, 3 rinses between washes) and blocked for 1 h in PBST (PBS, 1% Triton X-100) with 10% foetal bovine serum. Primary antibodies were incubated in PBST overnight or for 2 nights at 4 °C. Brains were washed in PBST and secondary antibodies incubated for 2 h, followed by washes in PBST and sequentially embedding in 50% and 70% glycerol. Brains were mounted in a 1:1 mixture of 70% glycerol and Vectashield (Vector Laboratories). Antibodies used: rabbit anti-GFP (1:1000, kind gift from U. Mayor) (Gil-Ranedo et al, 2019), chicken anti-GFP (1:500, Millipore, 06-896), guinea pig anti-Dpn (1:2000, kind gift from J. Knoblich) (Levy and Larsen, 2013), rabbit anti-Ase (1:10,000, kind gift from Y. Jan) (Brand et al, 1993), mouse anti-pH3 (1:1000, Abcam, ab5176), rabbit anti-FBL (1:200, Abcam, ab5821), mouse anti-HEATR1 (1:500, Santa Cruz Biotechnology, sc-390445), guinea pig anti-d-Myc (1:100, kind gift from G. Morata) (Herranz et al, 2008). For TUNEL, brains were fixed as above and In Situ Cell Death Detection Kit, TMR red (Roche) used following manufacturer's instructions. For EU assays, dissected brains were incubated with 10 mM EU in PBS (1 h; 25 °C) and fixed as above. Primary antibodies were incubated overnight and washed. Incorporated EU was detected using Click-iT RNA Imaging Kit (Invitrogen) following manufacturer's guidelines. Secondary antibodies and DAPI were incubated and washed as described.

Human tissue immunohistochemistry was performed as described (Hilton et al, 2009). Briefly, paraffin tissue sections were de-waxed, rehydrated and endogenous peroxides blocked with 3% H$_2$O$_2$ in methanol (45 min). Antigen retrieval performed by heating sections in EDTA pH 9 for 30 min using a microwave. Sections were blocked with horse serum (Vector Laboratories) for 30 min. Mouse anti-HEATR1 (1:100, Santa Cruz Biotechnology, sc-390445) incubated overnight at room temperature (RT). Signal developed using Vectastain Universal Elite ABC kit (Vector Laboratories) and visualized with SigmaFast DAB tablets (Sigma). Counterstain with home-made Mayer´s haematoxylin for 1–2 min. IDH1 staining was performed using a Ventana BenchMark Ultra (Roche) automated slide stainer. Slides were pre-treated with ULTRA cell conditioner CC2 (Roche) and mouse anti-IDH1 R132H (1:50, Dianova, DIA-H09) incubated for 32 min at RT. Signal detected using OptiView DAB IHC Detection Kit (Roche). Results were also analysed and reviewed by a neuropathologist (DAH).

## Cell transfection

U87MG and U251MG cell transfection with esiRNAs used Lipofectamine 2000 (Invitrogen) following manufacturer's guidance. Briefly, $0.2 \times 10^6$ cells were transfected with 100 pmol of esiRNA in Opti-MEM (Gibco) and incubated for 18 h. Media was replaced with complete MEM with 10% FBS and cells incubated for 48 h before processing. For plasmid transfections, $0.1 \times 10^6$ HeLa, $0.2 \times 10^6$ U87MG or $0.6 \times 10^6$ 293T cells were seeded per M6 well and transfected as above with 2 μg or 1 μg (MYC-HA localisation assays) of pCMV-HA-N or pCMV-HA-h-c-Myc (gifts from S. Matsufuji) and/or 2 μg or 1 μg (RT-qPCRs using 293T cells) of pIRES-FLAG or pIRES-FLAG-HEATR1 (gifts from Z. Lou), and incubated for 7 h (HeLa; 293T) or 4 h (U87MG). Media was replaced by complete MEM with 10% FBS and cells incubated for 17 h (HeLa), 24 h (293T) or 4 h (U87MG) before processing. Cell transfections towards co-IPs are described below.

## Lentiviral production and transduction

Lentiviral particles were produced as described (Gil-Ranedo et al, 2011). Briefly, 293T cells were co-transfected in Opti-Mem with plasmids pLKO.1-puro (control, Sigma) or pLKO.1-puro carrying *HEATR1* shRNA (TRCN0000137322; TRCN0000136697; Sigma), psPAX2 (12260, Addgene) and pMD2.G (12259, Addgene) using Lipofectamine 2000 (Invitrogen) and incubated for 18 h. Medium was replaced with complete MEM with 10% FBS or DMEM/F12. Supernatants containing lentiviral particles collected after 54–56 h, filtered and used immediately or stored at −80 °C. For U87MG transduction, $0.2 \times 10^6$ cells were seeded per M6 well in 1.5 ml complete MEM with 10% FBS and incubated for 18 h. Medium was replaced with 1 ml of complete MEM with 10% FBS, 600 μL of lentiviral-containing medium and polybrene (4 μg/ml, Sigma) and cells incubated for 18 h. Medium was replaced by 2 ml complete MEM + 10% FBS. After 6 to 8 h incubation, cells were selected by adding puromycin (0.8 μg/ml) for 168 h before processing.

GSC transduction was performed as described (Gil-Ranedo et al, 2011). Briefly, $0.5 \times 10^6$ cells were seeded per M6 well in 1.5 ml Opti-MEM and incubated for 4 h. Medium was replaced with 1 ml of complete DMEM/F12, 600 μL of lentiviral-containing medium and polybrene (4 μg/ml, Sigma). Cells were incubated for 18 h and next transferred to T25 flasks with 6 ml of complete DMEM/F12. After 6 to 8 h incubation, cells were selected by adding puromycin (0.8 μg/ml) for 168 h before being processed.

## Immunofluorescence, TUNEL, EdU, EU and OPP assays in cells

Cells were grown on coverslips and stained as described (Barros et al, 2009; Gil-Ranedo et al, 2011) with minor modifications. For GSCs, coverslips were pre-treated with laminin (10 µg/ml, Sigma) for 2 h at 37 °C and washed in PBS. Briefly, GSC tumourspheres were dissociated into individual cells and attached in the laminin-coated coverslips overnight at 37 °C. Cells were sequentially rinsed in PBS, fixed (15 min in 4% formaldehyde/PBS), washed in PBS (2 × 5 min, 3 rinses between washes), permeabilised (15 min in PBS, 0.2% Triton X-100) and blocked (20 min in PBS, 0.1% Triton X-100, 1% FBS). Primary antibodies incubated in blocking buffer overnight at 4 °C. Following washes, secondary antibodies were incubated in blocking buffer for 2 h at RT and next washed. Coverslips with cells were mounted using ProLong Diamond antifade mountant with DAPI (Invitrogen). Antibodies used: rabbit anti-FBL (1:200, Abcam, ab5821), mouse anti-Ki-67 (1:75, Dako, M7240), mouse anti-UBF (1:500, Santa Cruz Biotechnology), mouse anti-HEATR1 (1:100, Santa Cruz Biotechnology, sc-13125), mouse anti-RPA194 (1:50, Santa Cruz Biotechnology, sc-48385), rat anti-HA (1:1000, Roche, 11867431001), mouse anti-c-Myc (1:300, Santa Cruz Biotechnology, sc-40), mouse anti-Nucleophosmin (NPM, B23, 1:100, Santa Cruz Biotechnology, sc-271737).

For TUNEL, cells were fixed as described above and In Situ Cell Death Detection Kit, TMR red (Roche) used following manufacturer's instructions. For EdU incorporation, U87MG and U251MG cells were incubated with 10 µM EdU/MEM (1 h 37 °C). GSCs were incubated with 20 µM EdU/complete DMEM/F12 (1 h 37 °C) and next seeded on laminin-coated coverslips for 20 min at 37 °C, washed in PBS and fixed as above. Incorporated EdU detected using Click-iT EdU Imaging kit following manufacturer's instructions (Invitrogen). For RNA synthesis assays, cells were incubated with 1 mM EU/MEM (1 h RT) and fixed as above. Incorporated EU detected using Click-iT RNA Imaging Kit (Invitrogen) following manufacturer's guidelines. For OPP incorporation assays, cells were incubated with 20 µM OPP/MEM (30 min at 37 °C) and fixed as above. Incorporated OPP detected using Click-iT Plus OPP Protein Synthesis Assay Kit (Invitrogen) following manufacturer's instructions.

## Image acquisition and processing

Images were acquired on a Leica SP8 or Leica SPE confocal laser-scanning microscope, using LAS X software. Quantification in *Drosophila* larval brains used z-stacks of 1.5 µm step size (late stages), 1 µm or 0.5 µm (early stages), comprising whole brain lobes. All representative images are single optical sections except HEATR1 and MYC localisation and EU incorporation analyses in U87MG cells, in which projections of z-stacks 0.3 µm step size were used encompassing whole cells. *Drosophila* brain images are shown anterior up. Images were processed in Fiji v2.0 or Adobe Photoshop CS6 and figures assembled in Adobe Illustrator CS6.

## Co-immunoprecipitations and western blotting

For co-immunoprecipitations, 293T cells transfected with 0.5 µg pCMV-HA-N or pCMV-HA-h-c-Myc, and/or 1 µg of pIRES-FLAG or pIRES-FLAG-HEATR1 were lysed 24hpt in lysis buffer (20 mM Tris-HCl pH 7.5, 150 mM NaCl, 1 mM EDTA, 1% NP-40, 0.5%

sodium deoxycholate; pH 7.9) with protease inhibitor (Complete, EDTA-free; Sigma) and phosphatase inhibitors (cocktails B and C; Santa Cruz Biotechnology). Lysates were spun (14,000 rpm, 30 min, 4 °C), supernatants quantified (BCA protein assay, Pierce) and pre-cleared (1 h, 4 °C, rotating) with 50 µl Dynabeads Protein G (Invitrogen) followed by incubation (1 h, 4 °C, rotating) with 50 µl Dynabeads Protein G (Invitrogen) bound to 1 µg of anti-Flag M2 (Sigma, F3165) or 1 µg of anti-HA (3F10; Roche, 11867431001) antibodies, following manufacturer's guides. Samples were washed 4 times with lysis buffer and eluted as indicated by the manufacturer. Proteins detected by SDS-PAGE and western blotting using standard procedures.

Protein extraction from tissue samples used lysis buffer (25 mM Tris, 0.15 M NaCl, 1 mM EDTA, 1% NP-40, 5% glycerol; pH 7.4; 1 ml per 10 mg tissue) with protease (Complete, EDTA-free; Sigma) and phosphatase (cocktails B, C; Santa Cruz Biotechnology) inhibitors and placed at −80 °C overnight. Tissues were next dissociated using plastic grinders followed by pestle-mortars, samples spun (14,000 rpm, 30 min, 4 °C) and supernatants quantified (BCA protein assay, Pierce).

Protein cell lysates were performed as described (Gil-Ranedo et al, 2019). Briefly, cells were lysed in lysis buffer (30 min on ice), spun (14,000 rpm, 30 min, 4 °C) and protein quantified as above. When indicated, commercially available human brain tissue lysate (Abcam; ab29466) was used as non-tumour control. SDS-PAGE and western blotting were performed using standard procedures. Antibodies used: rabbit anti-FBL (1:500, Abcam, ab5821), mouse anti-HEATR1 (1:500, Santa Cruz Biotechnology, sc-390445), mouse anti-β-Actin (1:10,000, CST, 4967), mouse anti-TRIM3 (27; 1:500, Santa Cruz Biotechnology, sc-136363), mouse anti-RPA194 (1:500, Santa Cruz Biotechnology, sc-48385), mouse anti-c-Myc (9E10; 1:500, Santa Cruz Technology, sc-40), rabbit anti-c-Myc (1:500, CST, 5605), mouse anti-p53 (1:500, Santa Cruz Biotechnology, sc-126), mouse anti-α-Tubulin (1:1000 Sigma, T5168), mouse anti-UBF (1:100, Santa Cruz Biotechnology, sc-13125), mouse anti-Nucleophosmin (NPM, B23; 1:100, Santa Cruz Biotechnologies, sc-271737), mouse anti-Flag M2 (Sigma, F3165), rat anti-HA (3F10; Roche, 11867431001).

## Flow cytometry

Apoptosis analysis performed using Annexin V apoptosis detection kit (Invitrogen) following manufacturer's guidelines. Briefly, cells were washed, resuspended in 200 µL in of binding buffer with 5 µL of Annexin V-FITC and incubated (15 min, RT). Cells were next washed, resuspended in 200 µL of binding buffer with 5 µL of propidium iodide and incubated (15 min, RT) and kept on ice. Samples were analysed on a FACSAria II Flow Cytometer (BD Biosciences) using 488 nm and 633 nm laser lines after compensation using appropriate controls. BD FACSDiva (v6.1.3, BD Biosciences) and FlowJo (vX.0.7, TreeStar) software were used for data acquisition and analysis, respectively.

## Soft agar colony assays

GSC soft agar colony formation assays were performed as described (Gil-Ranedo et al, 2011). Briefly, 2 ml of complete DMEM/F12 medium with 0.35% agar were added into M6 wells and $1 \times 10^4$ (GSC-5) or $5 \times 10^3$ (GSC-8) puromycin-selected cells seeded on top in another 2 ml of complete DMEM/F12 medium with 0.5% agar.

GSC-5 and GSC-8 cells were incubated for 9 and 10 days, respectively. Colony size and morphology were recorded with a Leica DM IL LED microscope coupled to a Leica DFC3000 G camera. For colony numbers, 1 ml per well of complete DMEM/F12 medium containing 600 µg of MTT (Sigma) was added overnight to allow scoring at lower magnification towards scoring. Imaging was conducted on a Leica DM1000 LED microscope coupled to a Leica MC170 HD camera. Colony number scored using nucleus counter plugin included in Fiji/ImageJ (Schindelin et al, 2012).

### Data quantification and statistical analysis

Transcriptome data pre-processing and normalisation were carried out in R with additional Bioconductor packages. Background correction method normexp implemented in the backgroundCorrect function of the limma package was applied with an offset of 50 (Ritchie et al, 2015). Spots flagged by the scanner software or corresponding to spikes were removed. Lowly expressed genes were removed from further analysis by filtering genes with signals less than 100 in more than half of the samples (1402 genes remaining). To balance the intensity of both signal channels and remove potential dye bias, the optimised intensity normalisation method (OIN) of the OLIN package for normalisation of two-colour array data (Futschik and Crompton, 2005) was applied. Additional quantile normalisation was used to scale signal intensities across arrays. The log ratios between *brat* INPs (*brat* TICs) and control iINPs were based on reconstituted single channel expression values after intra-microarray and inter-microarray normalization via OIN and quantile normalisation (1211 unique genes remaining). The statistical significance of differential expression values was calculated using the LPEadj package (Murie and Nadon, 2008). *P*-values were corrected for multiple testing and converted into false discovery rates (FDR) using the resampling method implemented in LPEadj package. Genes with FDR < 0.1 (358) were considered for further analysis, including expression validation. RT-qPCR gene expression analysis was quantified using the Livak method (Livak and Schmittgen, 2001).

Quantifications of individual cell or GSC colony sizes were calculated measuring maximum cell or colony diameters, respectively (Chell and Brand, 2010; Gil-Ranedo et al, 2019).

Relative nucleoli sizes in cultured cells are ratios between nucleolar (marked by UBF, FBL and NPM1 expression) and nuclear (DAPI) areas. Relative Pol I area is the ratio of RPA194-labelled and nuclear areas. In *Drosophila* brain cells, relative nucleoli sizes are ratios between nucleoli (FBL) and maximum cell diameters, and relative EU signal is the product of EU mean intensity and EU maximum areas (integrated densities). Only NSC progeny was scored in EU assays. In cultured cells, relative OPP signal is provided as ratio between OPP signal pixel intensity and number of cells per field. HEATR1 and MYC nucleolar localisation enrichment, and nuclear and nucleolar EU incorporation, were calculated using HEATR1, MYC or EU and FBL signal profiles across a line over the nucleus (DAPI) of each measured cell, crossing at least one nucleolus labelled by FBL antibodies. Plotted signals were calculated as rolling averages of 0.36 µm. Similarly, in *Drosophila* brain cells, relative MYC expression was calculated using MYC signal profiles across a single line over each scored cell and crossing at least one nucleolus labelled with FBL. Relative total MYC was calculated averaging the signal across the whole profile. Relative nucleolar/total MYC ratios used the MYC nucleolar signal

overlapping with FBL. Cells scored were located at similar depths within brain lobes of strains analysed.

Statistical analyses were performed using SigmaPlot Version 12.5 (Systat Software) or Prism 8.2 (GraphPad Software). For correlation analysis, the Pearson´s correlation coefficient ($r$) between *TRIM3* and *HEATR1* expression was calculated. Shapiro–Wilk tests were used to evaluate data normality. Unpaired two-tailed *t*-tests were applied on data for which the Shapiro–Wilk tests indicated normality. Mann–Whitney tests were used on data for which the Shapiro–Wilk tests rejected normality. $p < 0.05$ considered statistical significant. For RT-qPCR data analysis in Figs. 1E and EV1D, Holm correction was applied on $p$ values following Shapiro–Wilk tests and unpaired two-tailed *t*-tests to control for family-wise error rate (FWER) in sample groups compared. Sample numbers (n) are indicated in figures or legends. Histograms show mean ± standard error of the mean (s.e.m). Box plots represent 25th and 75th percentiles, black central bands indicate medians, red or white central bands specify means, whiskers indicate 10th and 90th percentiles. *Drosophila* data were obtained from at least 4 brains lobes from 4 different brains. Data from in vitro assays derive from a minimum of three biological replicates (three separate culture and subsequent processing). Sample numbers and replicas were sufficient to detect statistical significances between conditions tested. No samples were excluded unless procedures failed. Blinding was not performed. Data quantification as described above were applied equally in all conditions.

## Data availability

Raw and pre-processed transcriptome data generated in this study are deposited in the NCBI Gene Expression Omnibus (https://www.ncbi.nlm.nih.gov/geo/) under the Series record GSE190133. See Dataset EV1 for processed transcriptome data.

## Peer review information

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

## Acknowledgements

We are grateful to FlyChip (University of Cambridge, UK) and the Plymouth light microscope service (PLiMS) core facility (University of Plymouth, UK) for technical support. We acknowledge the Kyoto *Drosophila* Genetic Resource Consortium, the Vienna *Drosophila* Resource Center and the Bloomington *Drosophila* Stock Center for *Drosophila* strains, and the Glioma Cellular Genetics Resource (University of Edinburgh, UK) for the NSC line used. Tissue samples were obtained from the Department of Neuropathology, University Hospitals Plymouth as part of the UK Brain Archive Information Network (BRAIN UK; License 14/004), funded by the Medical Research Council UK and Brain Tumour Research. We are extremely grateful to the patients who agreed to tissue donation. We are very thankful to those that kindly provided antibodies, *Drosophila* lines and plasmids (see Methods). We also thank H. Robinson for initial *Drosophila* crosses leading to *brat* and *pnt-gal4* combined lines and T. Madgett, B. Fisher and S. Russell for helpful discussions. This work was sponsored by Tenovus Cancer Care (PhD2011/L39), Brain Research UK (201617-05), Brain Tumour Research, FCT (Fundação para a Ciência e Tecnologia; XPL/CCI-BIO/1650/2021, UIDB/04539/2020, UIDP/04539/2020 and LA/P/0058/2020 to MEF), School of Biomedical Sciences and Peninsula Medical School, University of Plymouth, UK.

## Author contributions

**Laura R Diaz**: Formal analysis; Investigation; Visualization; Methodology. **Jon Gil-Ranedo**: Conceptualization; Formal analysis; Supervision; Investigation; Visualization; Methodology; Writing—original draft; Writing—review and editing. **Karolina J Jaworek**: Formal analysis; Investigation; Methodology; Writing—review and editing. **Nsikan Nsek**: Formal analysis; Investigation; Methodology. **Joao Pinheiro Marques**: Formal analysis; Investigation; Methodology. **Eleni Costa**: Formal analysis; Investigation; Methodology. **David A Hilton**: Resources; Formal analysis; Investigation; Methodology. **Hubert Bieluczyk**: Formal analysis; Investigation. **Oliver Warrington**: Formal analysis; Investigation. **C Oliver Hanemann**: Supervision; Funding acquisition; Writing—review and editing. **Matthias E Futschik**: Formal analysis; Investigation; Methodology; Writing—review and editing. **Torsten Bossing**: Conceptualization; Resources; Formal analysis; Supervision; Investigation; Methodology; Writing—original draft; Writing—review and editing. **Claudia S Barros**: Conceptualization; Resources; Formal analysis; Supervision; Funding acquisition; Investigation; Visualization; Methodology; Writing—original draft; Project administration; Writing—review and editing.

## Disclosure and competing interests statement

The authors declare no competing interests.

# Expanded View Figures

**Figure EV1.  Expression and ontology analysis of human orthologues of identified genes in *brat* TIC transcriptomics.**

(A) Percentage of grade II, III and IV gliomas with *TRIM3* homozygous or hemizygous deletions. Biological replicates: 226 (grade II); 244 (grade III); 510 (grade IV). (B) RT-qPCR analysis of *TRIM3* in grade II Diffuse Astrocytoma (DA), GBM and non-tumoral control brain samples. Biological replicates: 5–19. Unpaired two-tailed t-tests. Box plot represent 25th and 75th percentiles, central black bands indicate medians, central red bands specify means, whiskers indicate 10th and 90th percentiles. (C) Immunoblots of TRIM3 in GSCs (GSC-5). Signal in mouse cerebellum shown as positive control. β-Actin: loading control. (D) RT-qPCRs of a subset of identified genes in grade II DA and GBM (upper panel), or in GSCs (lower panel), versus non-tumoral brain tissue (Fold Change, FC). Error bars: s.e.m. Biological replicates: 5–19 (upper panel), 3–9 (lower panel). Technical replicates: 1–3 (lower panel). Unpaired two-tailed t-tests followed by Holm correction. (E) GSEA of human orthologues of genes differentially expressed in *brat* TICs versus control iINPs (single best matches of dataset FDR < 0.1; DIOPT score ≥ 5) using as ranked list the gene dataset from the GSC genome-wide CRISPR screening ordered by Bayes Factor (BF) (MacLeod et al, 2019). Higher BF values indicate greater confidence in GSC fitness reduction after CRISPR-Cas9 gene knockout. NES, normalized enrichment score. Empirical p-value estimated based on $10^7$ random permutations. (F) Overrepresented KEGG pathways in identified human orthologues dataset (DIOPT score ≥ 5). Pathways also grouped in broader categories. Data information: ***$p \leq 0.001$; **$p \leq 0.01$; *$p \leq 0.05$; $p > 0.05$, ns (non-significant).

◀

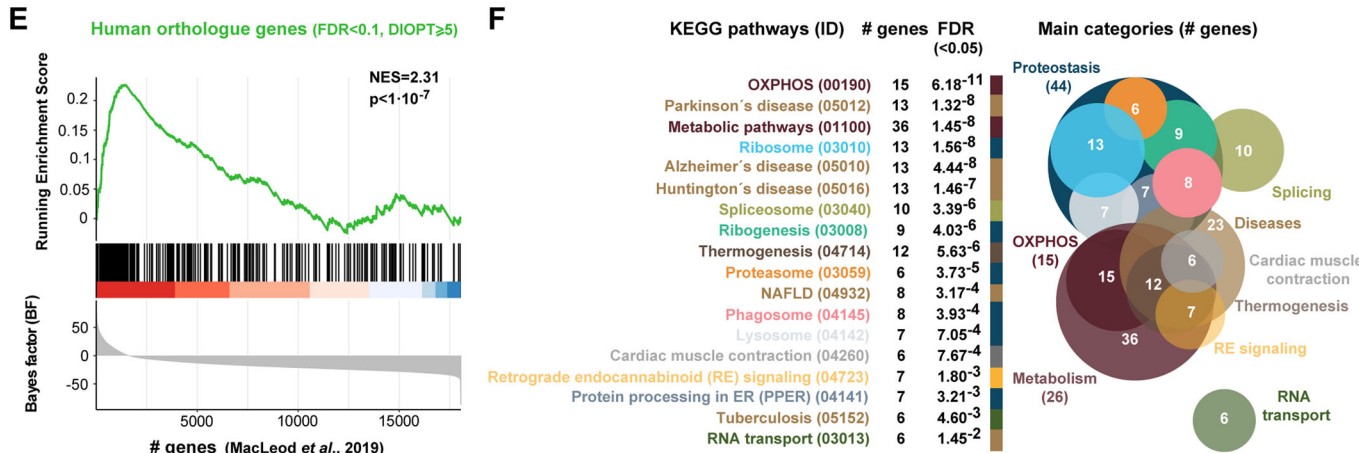

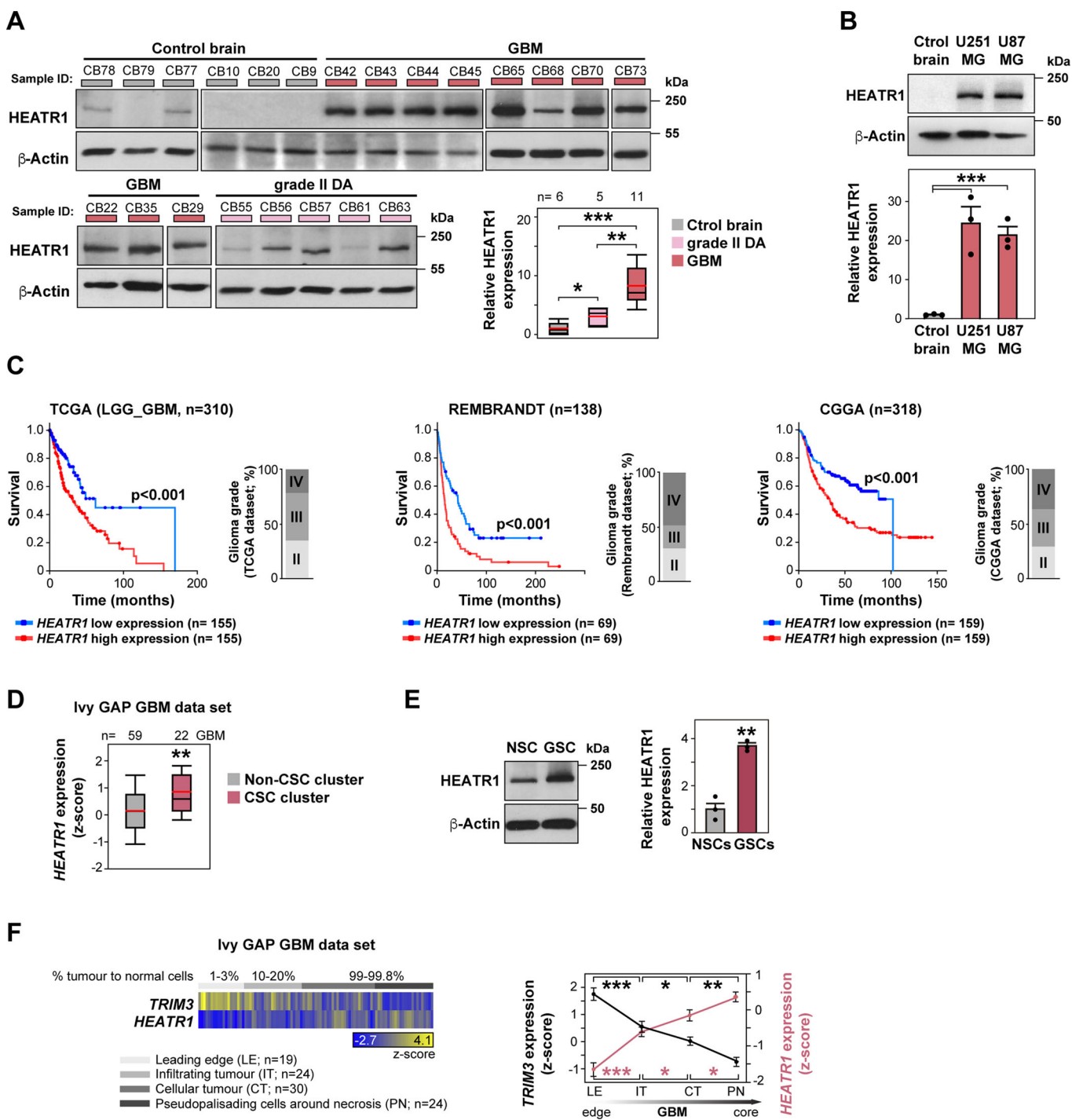

**Figure EV2. HEATR1 expression is increased in glioma and inversely correlates with patient survival.**

(**A, B**) Immunoblots of HEATR1 in grade II DA, GBM and non-tumoral control (Ctrl) brain samples (**A**; biological replicates: 5–11), as well as in U251MG, U87MG GBM cell lines and control brain (**B**; biological replicates: 3). β-Actin: loading control. Error bars: s.e.m. Quantification of HEATR1 signal. Unpaired two-tailed t-tests. (**C**) Relative patient survival with the 25% higher (red) or 25% lower (blue) *HEATR1* expression in glioma (grades II to IV). Biological replicates: 310 (TCGA); 138 (REMBRANDT); 318 (CGGA). Log-Rank *p*-values. (**D**) *HEATR1* expression in putative cancer stem cell (CSC) and non-CSC clusters in GBM. Clusters identified by expression of 17 reference probes via in situ hybridization, Ivy GAP. Biological replicates: 22–59. Unpaired two-tailed t-test. (**E**) HEATR1 expression analysis by immunoblotting in GSCs (GSC-5) versus NSCs (biological replicates: 3). β-Actin: loading control; Error bars: s.e.m.; Unpaired two-tailed t-tests. (**F**) *TRIM3* and *HEATR1* relative expression levels in four GBM regions indicated. Tumour to normal cell ratios (%). Sample numbers for each region: Leading edge, 19; Infiltrating tumour, 24; Cellular tumour, 30; Pseudopalisading cells around necrosis, 24. RNAseq data, IVY GAP. Unpaired two-tailed t-tests. Error bars: s.e.m. Data information: Box plots represent 25th and 75th percentiles, central black bands indicate medians, central red bands specify means, whiskers indicate 10th and 90th percentiles. ***p ≤ 0.001; **p ≤ 0.01; *p ≤ 0.05; p > 0.05.

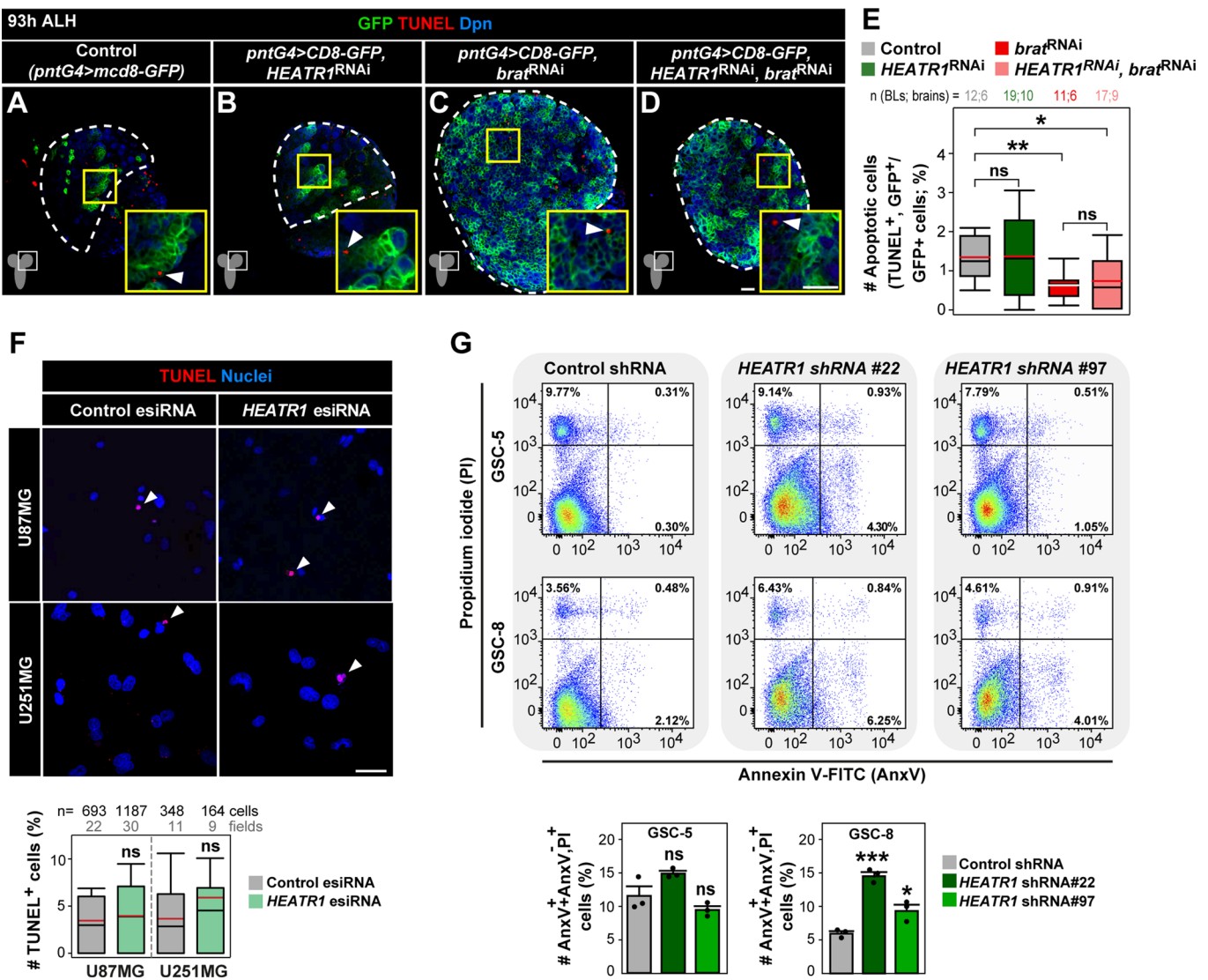

**Figure EV3.** *HEATR1* inhibition has no impact in *brat*-deficient tumour, control brain or GBM cell death but exerts a mild effect on GSCs.

(A–E) TUNEL labelling and immunostainings of GFP and Dpn in type II NSC lineages expressing *CD8-GFP* (no-tumour control) and *HEATR1*RNAi, *brat*RNAi (tumour) or *HEATR1*RNAi, *brat*RNAi (*HEATR1*-deficient tumour) at 93 h ALH. TUNEL+GFP+ quantification (% of GFP+ cells, biological replicates: 11–19, unpaired two-tailed t-tests; E). Dashed lines: central brain region; Insets: higher magnifications. White arrowheads: TUNEL+ cells. Scale bars: 10 μm. (F) TUNEL labelling of U87MG and U251MG cells 48 hpt with *HEATR1*-esiRNA or control *GFP*-esiRNA. Nuclei (DAPI). Scale bar: 50 μm. Arrowheads: TUNEL+ cells. TUNEL+ cell quantification: % of DAPI+ cells; 22–30 (U87MG) and 9–11 cell images (fields) from 3 biological replicates; unpaired two-tailed t-tests. (G) Annexin-V-FITC (AnxV) and red fluorescent Propidium Iodide (PI) labelling in GSCs (GSC-5; GSC-8) by flow cytometry. Representative dot plots (% cells in gated subpopulations). Quantification of cell subpopulations: 12,000–50,000 cells per condition from 3 biological replicates, error bars: s.e.m., unpaired two-tailed t-tests. Data information: Box plots represent 25th and 75th percentiles, central black bands indicate medians, central red or white bands specify means, whiskers indicate 10th and 90th percentiles. ***$p \leq 0.001$; **$p \leq 0.01$; *$p \leq 0.05$; $p > 0.05$, ns (non-significant).

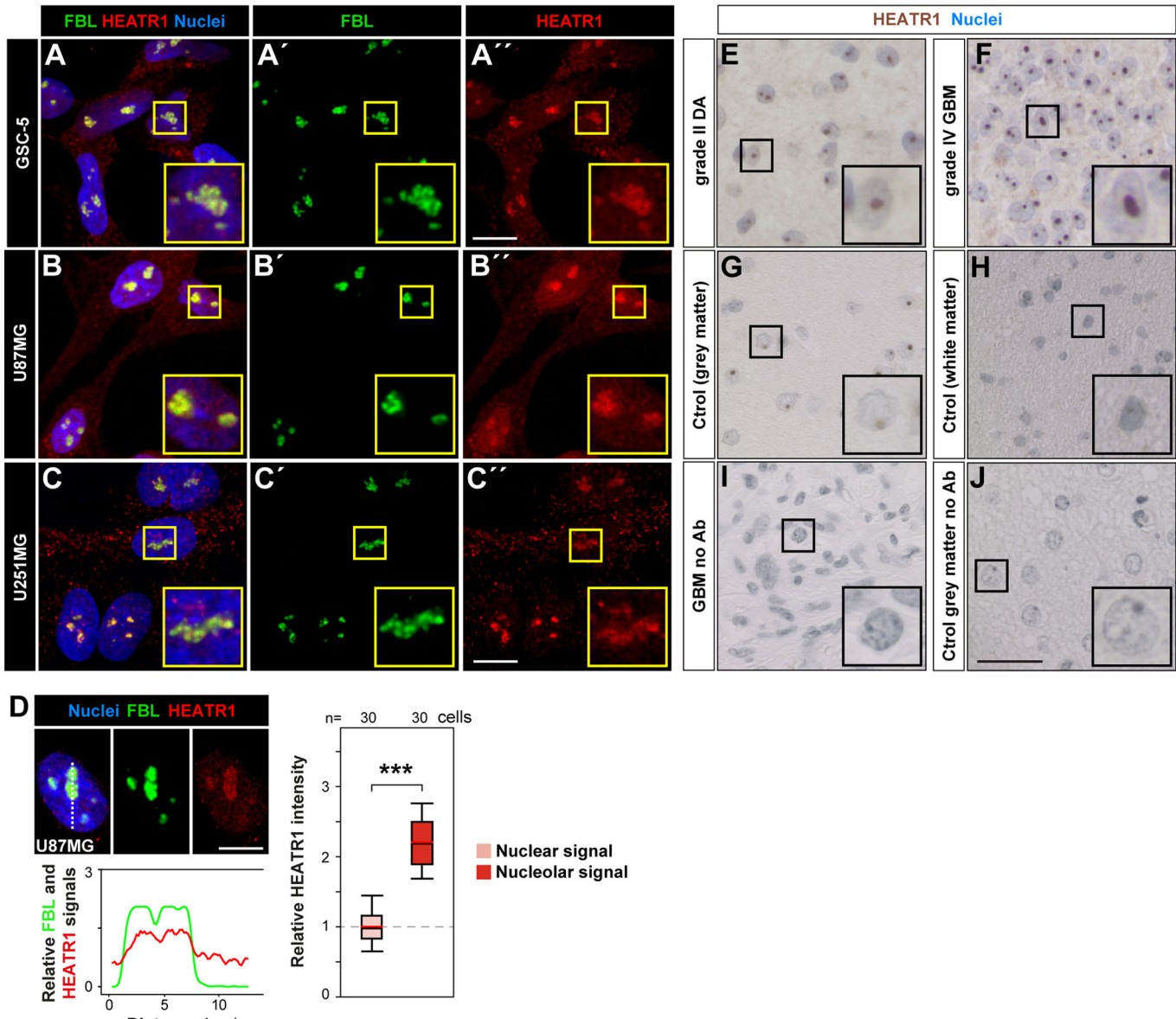

**Figure EV4.   HEATR1 localises predominantly in nucleoli of glioma cells.**

(**A–C"**) Immunostainings of HEATR1 and FBL in GSCs (GSC-5; **A–A"**) and GBM cell lines (U87MG, **B–B"**; U251MG, **C–C"**). Nuclei (DAPI). FBL (**A'–C'**) and HEATR1 (**A"–C"**) single channels in green and red, respectively. Insets: higher magnification. (**D**) Representative image and intensity profile plot for FBL and HEATR1 relative signals in nucleus and nucleoli regions (dashed line; left panels) of U87MG cells. Relative nucleolar to nuclei HEATR1 signal quantification, 30 cells scored from 3 biological replicates, unpaired two-tailed t-test (right panel). Box plot represent 25th and 75th percentiles, central black bands indicate medians, central red bands specify means, whiskers indicate 10th and 90th percentiles. ***$p \leq 0.001$. (**E–J**) Immunohistochemistry of HEATR1 in grade II DA ($n = 9$; **E**), GBM ($n = 9$; **F**) and control non-tumour brain samples in grey (**G**) and white (**H**) matter regions ($n = 10$). No primary antibody controls (**I, J**). Nuclei: haematoxylin counterstain. See Fig. EV2 for HEATR1 protein quantifications. Data information: Scale bars: 10 μm.

                        

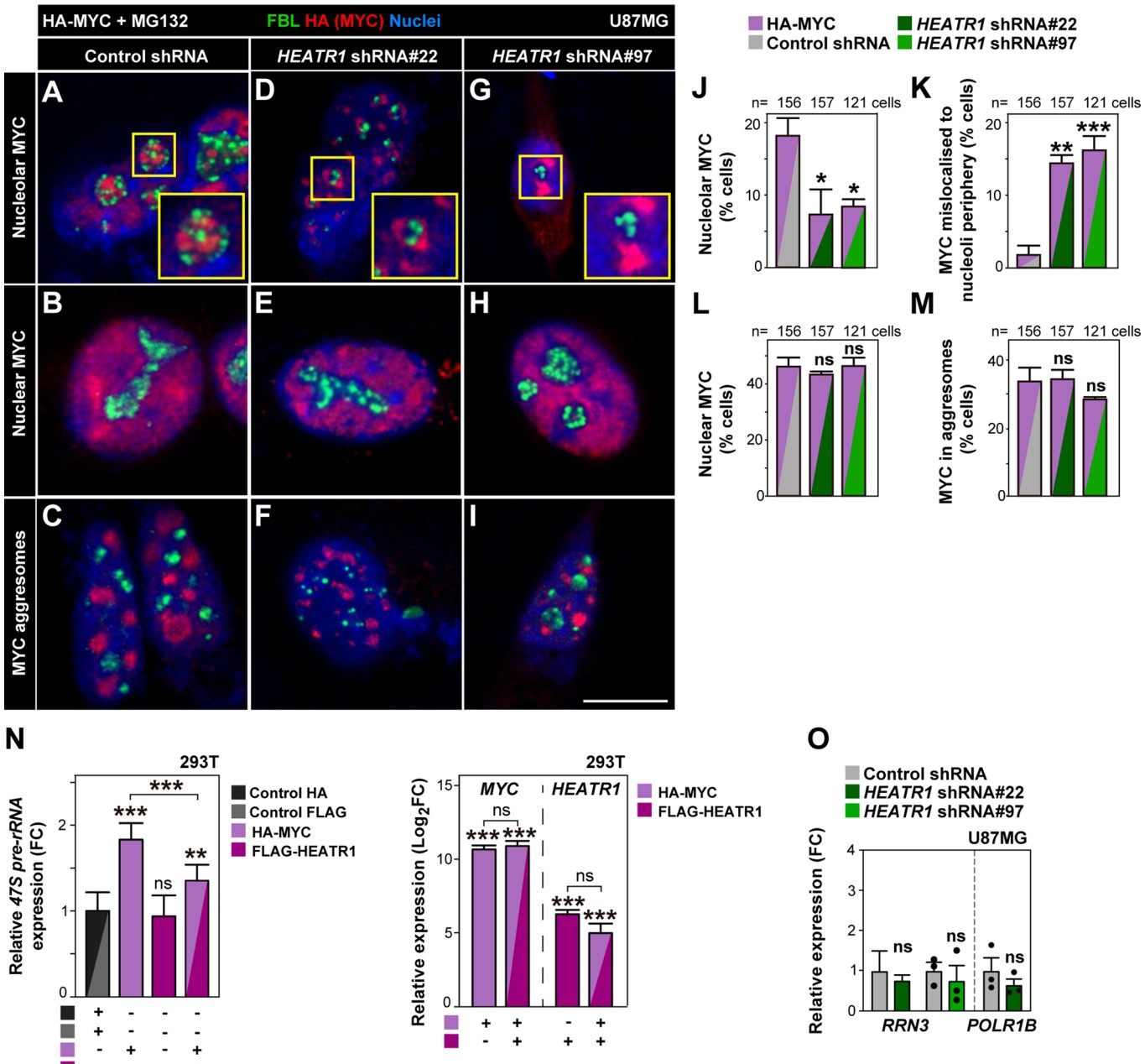

**Figure EV5. HEATR1 is required for exogenous MYC localisation in nucleoli.**

(A–M) Immunostainings of FBL and HA-MYC in U87MG GBM cells 168 hpi with *HEATR1*-shRNAs (D–I) or control shRNAs (A–C) and 6 h after transfection with HA-tagged full-length MYC (HA-MYC) plus MG132 treatment. Nuclei (DAPI). Insets: higher magnification. Scale bar: 10 µm. Quantification of cells (% of DAPI⁺) showing nucleolar MYC (J), mislocalised MYC into nucleolar periphery (K), nuclear MYC (L) and MYC in nuclear aggregates (aggresomes; M). Error bars: s.e.m. 121–157 cells scored from 3–4 biological replicates. Unpaired two-tailed t-tests. (N) RT-qPCR analysis of *47S* pre-RNA (left panel) in 293T cells transfected with HA-tagged MYC (HA-MYC) or control HA vector, and Flag-tagged HEATR1 (Flag-HEATR1) or control Flag vector. RT-qPCR analysis of *MYC* and *HEATR1* (right panel) in 293T cells expressing Flag-HEATR1 and/or HA-MYC versus controls. Fold change (FC). Biological replicates: 3, technical replicates: 3. Error bars: s.e.m. Unpaired two-tailed t-tests except Mann–Whitney test in *HEATR1* analysis on double HEATR1-FLAG, HA-MYC expressing samples. (O) RT-qPCR analysis of *RRN3* and *POLR1B* (fold change, FC) in U87MG cells 168 hpi with *HEATR1*-shRNAs versus control shRNAs. Fold change (FC). Biological replicates: 3; technical replicates: 2 (*RRN3* upon *HEATR1* shRNA# 22 versus control). Error bars: s.e.m. Unpaired two-tailed t-tests except *RRN3* analysis upon *HEATR1* shRNA# 22 versus control (Mann–Whitney test). Data information: ***$p \leq 0.001$; **$p \leq 0.01$; *$p \leq 0.05$; $p > 0.05$, ns (non-significant).

