## [Peer Review File · EMBO Reports]

Ribogenesis boosts controlled by HEATR1-MYC interplay promote transition into brain tumour growth

Laura Diaz, Jon Gil-Ranedo, Karolina Jaworek, Nsikan Nsek, Joao Marques, Eleni Costa, David Hilton, Hubert Bieluczyk, Oliver Warrington, C. Oliver Hanemann, Matthias Futschik, Torsten Bossing, and Claudia Barros

DOI: [10.15252/embr.202356951](https://doi.org/10.15252/embr.202356951)

Corresponding author(s): *Claudia Barros (claudia.barros@plymouth.ac.uk)*

Review Timeline:

Submission Date:	6th Feb 23
Editorial Decision:	9th Mar 23
Revision Received:	8th Jun 23
Editorial Decision:	26th Oct 23
Revision Received:	16th Nov 23
Accepted:	22nd Nov 23

Editor: *Achim Breiling*

Transaction Report:

Dear Dr. Barros

Thank you for the transfer of your manuscript to EMBO reports. I have now received the reports from the three referees that were asked to evaluate your study, which can be found at the end of this message.

As you will see, the referees indicate that these findings are of interest. However, they have several comments, concerns, and suggestions, indicating that a major revision of the manuscript is necessary to allow publication of the study in EMBO reports. As the reports are below, and all the referee concerns need to be addressed as indicated in the reports, I will not detail them here.

Given the constructive referee comments, I would like to invite you to revise your manuscript with the understanding that all referee concerns must be addressed in the revised manuscript and in a detailed point-by-point response. Acceptance of your manuscript will depend on a positive outcome of a second round of review. It is EMBO reports policy to allow a single round of revision only and acceptance of the manuscript will therefore depend on the completeness of your responses included in the next, final version of the manuscript.

- 1) a .docx formatted version of the final manuscript text (including legends for main figures, EV figures and tables), but without the figures included. Figure legends should be compiled at the end of the manuscript text.
- 2) individual production quality figure files as .eps, .tif, .jpg (one file per figure), of main figures (up to 8) and EV figures. Please upload these as separate, individual files upon re-submission.

- 3) a complete author checklist, which you can download from our author guidelines (<https://www.embopress.org/page/journal/14693178/authorguide>). Please insert page numbers in the checklist to indicate where the requested information can be found in the manuscript. The completed author checklist will also be part of the RPF.

- 4) that primary datasets produced in this study (e.g. RNA-seq, ChIP-seq, structural and array data) are deposited in an appropriate public database. If no primary datasets have been deposited, please also state this in a dedicated section (e.g. 'No primary datasets have been generated and deposited'), see below.

The accession numbers and database should be listed in a formal "Data Availability" section (placed after Materials & Methods) that follows the model below. This is now mandatory (like the COI statement). Please note that the Data Availability Section is restricted to new primary data that are part of this study. This section is mandatory. As indicated above, if no primary datasets have been deposited, please state this in this section

Data availability

8) Regarding data quantification and statistics, please make sure that the number "n" for how many independent experiments were performed, their nature (biological versus technical replicates), the bars and error bars (e.g. SEM, SD) and the test used to calculate p-values is indicated in the respective figure legends (also for potential EV figures and all those in the final Appendix). Please also check that all the p-values are explained in the legend, and that these fit to those shown in the figure. Please provide statistical testing where applicable. Please avoid the phrase 'independent experiment', but clearly state if these were biological or technical replicates. Please also indicate (e.g. with n.s.) if testing was performed, but the differences are not significant. In case n=2, please show the data as separate datapoints without error bars and statistics. See also: <http://www.embopress.org/page/journal/14693178/authorguide#statisticalanalysis>

9) Please add scale bars of similar style and thickness to all the microscopic images, using clearly visible black or white bars (depending on the background). Please place these in the lower right corner of the images themselves. Please do not write on or near the bars in the image but define the size in the respective figure legend.

10) Please also note our reference format:

12) We now use CRediT to specify the contributions of each author in the journal submission system. CRediT replaces the author contribution section. Please use the free text box to provide more detailed descriptions and remove the author contributions from the manuscript. See also guide to authors:

<https://www.embopress.org/page/journal/14693178/authorguide#authorshipguidelines>

Please order the manuscript sections like this, using these names:

Title page - Abstract - Keywords - Introduction - Results - Discussion - Materials and Methods - Data availability section - Acknowledgements - Disclosure and Competing Interests Statement - References - Figure legends - Expanded View Figure legends

Finally, please note that all corresponding authors are required to supply an ORCID ID for their name upon submission of a revised manuscript. Please find instructions on how to link the ORCID ID to the account in our manuscript tracking system in our Author guidelines: <http://www.embopress.org/page/journal/14693178/authorguide#authorshipguidelines>

I look forward to seeing a revised version of your manuscript when it is ready. Please let me know if you have questions or comments regarding the revision.

Yours sincerely,

Referee #1:

The submitted manuscript reports an interesting study on the mechanisms that drive the transition from "normal" to "tumoral" cells in a simple *Drosophila* brain tumor type called brat. The authors conclude that at the earliest stages of brat neoplastic transformation HEATR1 becomes upregulated, hence enhancing MYC-dependent ribogenesis, which in turn promotes the transition from normal intermediate neural progenitors (INPs) to tumor initiating cells (TICs) and, eventually, leads to tumour growth. The provided evidence substantiates this conclusion.

The main asset of this manuscript is the comparative single-cell analysis that reveals a significant reshaping of the transcriptome in the transition from brat INPs to TICs already at a time that is well before the onset of tumor growth. The reported early TIC transcriptome and the results of the corresponding KEGG pathway enrichment data analysis presented in the manuscript are interesting even if somewhat confirmatory of the generally accepted key role of ribogenesis in tumor growth.

The justification for the parallel studies on glioma cells is feeble. There seems to be an urge to identify parallelisms between the fly tumor and some human counterpart. However, resemblance between fly brat tumours and human grade IV glioma is marginal while major differences are many.

Out of the many transcripts that belong to the early brat TICs' signature identified in this work, the authors chose HEATR1 for further in-depth analysis. They conclude that HEATR1 is required for brat TICs to undergo tumorous growth. Evidence substantiating this conclusion is significant. However, it does not discard a simpler, alternative interpretation: rather than being specifically required for brat TICs tumour growth, HEATR is generally required for growth, but its effect is only measurable at high growth rates. Indeed, as shown in Figure 2, HEATR1 loss has no effect on control cells when they proliferate at low rates (2 and 10 "# of dividing cells", Figure 2 Q and R) but has visible effects when control cells proliferate at a higher rate (20 "# of dividing cells", Figure 2 S). This alternative interpretation is consistent with a report that identifies *heatr1* and *myc* as the highest scoring genes in a high-throughput assay for directly monitoring nucleolar rRNA biogenesis (Bryant et al. *Open Biol.* 12: 210305. <https://doi.org/10.1098/rsob.210305>), which should be cited and discussed. The authors should prove this alternative interpretation wrong if they wish to maintain their claim.

Related to the part on GBM, statements like "GBM cell proliferation and tumorigenic potential depend on HEATR1" are void if such a dependence is unspecific. All tumours are dependent on most housekeeping genes. Please qualify the claimed relevance.

The manuscript also shows that nucleoli are larger in brat TICs than in control INPs even before TICs start to over-proliferate, which provides indirect, but suggestive data supporting a causative link between HEATR1 and enhanced ribogenesis.

Altogether this is a valuable effort to investigate the mechanisms driving the early stages of TICs. Taking advantage of refined transcriptome analysis from well-defined single-cell types the manuscript reports interesting insight on genes that may play key roles in the earliest stages of brat tumorigenesis. I find this manuscript acceptable for publication once the authors address the few concerns that I have raised.

Other issues

-A FDR < 0.15 is on the low significance side. I would suggest using a more stringent cut-off.

-"MYC expression in nucleoli (Fig 7A-D)" should be "MYC localisation in nucleoli (Fig 7A-D)"

Referee #2:

In this interesting paper, titled "Ribogenesis boost controlled by HEATR1-MYC interplay promotes transition into brain tumour growth", Diaz et al propose a role for increased ribogenesis in promoting growth in brain tumour initiating cells. They utilise the *Drosophila* larval brain as an *in vivo* model and patient derived glioblastoma stem cells as an *in vitro* model.

Their study starts from an impressive experiment whereby single tumour initiating cells are manually picked from the developing *Drosophila* brain, and their single-cell transcriptomes analysed. They uncover an evolutionary conserved role in brain tumorigenesis for HEATR1, which is found to be upregulated in a tumour background promoting (preceding) growth. HEATR1 is in addition found to associate with the transcription factor MYC (known to be involved in promoting tumour growth in these contexts), and promote its nucleolar localisation and boosting ribogenesis.

The manuscript and the conclusions are interesting, the experiments are overall well conducted, with different approaches and model systems supporting similar conclusions. The text is well written, and the figures are clearly presented. We are supportive of publishing this manuscript, but several minor concerns remain to be addressed, as listed below.

1. Figure EV1E requires statistical analysis.
2. All quantifications of mitosis in Figure 2 and 6 are absolute numbers, without normalisation. PH3 should be normalised to the total number of GFP+ Dpn+ cells, to obtain mitotic index as an indication for proliferation.
3. All experiments in *Drosophila* are conducted with HEATR1 loss-of-function, while an overexpression experiment in type II NBs or INPs seems relevant too, to study whether this can promote tumorigenesis. In particular given the clinical phenotype, whereby increased HEATR1 correlates with decreased survival. If a transgenic line is available, this is an obvious experiment that should be included. If not, I feel that generating a new line goes beyond the scope of the revision, and would not change the major conclusions.
4. If I understand correctly, in the *Drosophila* work, pntG4>CD8-GFP has been used as a control while the experiments have been conducted with RNAi. Ideally, a non-targeting RNAi from the same library should be included as control. I do not think that all fly experiments need to be repeated, in particular because of the robust phenotypes observed, and the similar observations from different approaches and model systems. However, the findings from the nucleolar analysis in Figure 5P-X are rather subtle, might be related to the different genetic background (control vs KK and TRIP lines used), and would benefit from including an appropriate control.

Editorial suggestions:

1. The structure of the manuscript is sometimes hard to follow. Initially, *Drosophila* work is followed by GBM work, but this is not consistent throughout the manuscript, which can be a bit confusing.
2. In the introduction, some parts would better fit in the discussion section eg the first paragraph of page 4.
3. Figure 2T,U: label the figure referring to 24h ALH.

Referee #3:

The manuscript by Diaz et al. seeks to understand the gene expression and cellular changes that occur early in cancer development. Based on microarray-based gene expression studies carried out using isolated single INP cells from control and brat mutant brains in *Drosophila*, the authors focused on the function of the HEATR1 and Myc genes/proteins in regulating cell growth and proliferation. While Myc is long known to regulate rRNA production (and increased cellular biomass), the authors link Myc and HEATR1 to the growth and division of TICs. It is also notable that the changes observed occur prior to the excessive proliferation that is seen when Brat is lost. Overall, this manuscript provides interesting insights using both *Drosophila* and mammalian systems.

I have only minor concerns, as listed below.

- Clarification is needed about the single cell microarray studies. The PCA plot implies that only 3 individual nuclei examined for control/brat. Is this correct? And then, were the confirmation RT-PCRs performed on independently isolated nuclei?
- It would have been nice to have a brief justification of why c-Myc was studied in the mammalian system when N-Myc may have been more biologically relevant.
- In Figure 1H, the String/cytoscape interaction plot shows both mitochondrial and cytoplasmic ribosomal proteins. It wasn't clear to me why mtRps were included here.
- In figure 6, *Drosophila* Myc seems to be sometimes in the nucleus and sometimes dispersed in- and outside the nucleus. Is this expected? In this figure, it is extremely difficult to discern any difference between Q' and R'. This (to me) can only be seen in Q' and R'.

- On p13, the authors may want to clarify that Myc binds (weakly) to rDNA in mammalian cells. In *Drosophila*, as far as I am aware, the activation of rRNA is indirect (Grewal et al. 2005).
- In the model presented, Myc and HEATR1 are shown as being regulated independently of each other. The HEATR1 gene has a canonical Myc-binding Ebox in its promoter region (CACGTG). Is it a direct target of Myc?

Minor:

- In several figures (at least 2, 5, & 6) panels are referred to out of order in the text.
- P10 "heatR1" should be "HEATR1"
- There seems to be a missing reference in introduction - related to the statement that restoring Brat 24hrs after depletion can prevent tumour growth where it cannot after 48 hours.
- It would be valuable to add gene labels to key genes in the volcano plot shown in figure 1C.

Manuscript: EMBOR-2023-56951-T

Diaz et al. *Ribogenesis boost controlled by HEATR1-MYC interplay promotes transition into brain tumour growth*

Response to reviewers' comments

Reviewer #1:

The submitted manuscript reports an interesting study on the mechanisms that drive the transition from "normal" to "tumoral" cells in a simple Drosophila brain tumor type called brat. The authors conclude that at the earliest stages of brat neoplastic transformation HEATR1 becomes upregulated, hence enhancing MYC-dependent ribogenesis, which in turn promotes the transition from normal intermediate neural progenitors (INPs) to tumor initiating cells (TICs) and, eventually, leads to tumour growth. The provided evidence substantiates this conclusion.

The main asset of this manuscript is the comparative single-cell analysis that reveals a significant reshaping of the transcriptome in the transition from brat INPs to TICs already at time that is well before the onset of tumor growth. The reported early TIC transcriptome and the results of the corresponding KEGG pathway enrichment data analysis presented in the manuscript are interesting even if somewhat confirmatory of the generally accepted key role of ribogenesis in tumor growth.

We thank the reviewer for the comments, acknowledging our study as interesting and our single cell analysis of cells transitioning into tumour growth as an asset. Our response to the concerns raised is below.

The justification for the parallel studies on glioma cells is feeble. There seems to be an urge to identify parallelisms between the fly tumor and some human counterpart. However, resemblance between fly brat tumours and human grade IV glioma is marginal while major differences are many.

We had no intention to exacerbate parallelisms and agree with the reviewer in that there are many differences between fly *brat* tumours and human grade IV glioma. We have therefore added a section to the manuscript (page 6, second paragraph) acknowledging the many chief species-specific differences and the need for focus on the study of specific and appropriate tumour aspects when using *Drosophila* brain tumour models, and discoveries probed in mammalian systems. We also highlight the use of the *brat* model allowing us a rare opportunity for the identification of cellular changes upon transition into tumour growth and why we sought to probe findings in human glioma stem cells and tissues. We have in addition toned down text referring to parallelisms in findings with the different model systems used.

*Out of the many transcripts that belong to the early brat TICs' signature identified in this work, the authors chose HEATR1 for further in-depth analysis. They conclude that HEATR1 is required for brat TICs to undergo tumorous growth. Evidence substantiating this conclusion is significant. However, it does not discard a simpler, alternative interpretation: rather than being specifically required for brat TICs tumour growth, HEATR is generally required for growth, but its effect is only measurable at high growth rates. Indeed, as shown in Figure 2, HEATR1 loss has no effect on control cells when they proliferate at low rates (2 and 10 "# of dividing cells", Figure 2 Q and R) but has visible effects when control cells proliferate at a higher rate (20 "# of dividing cells", Figure 2 S). This alternative interpretation is consistent with a report that identifies *heatr1* and *myc* as the highest scoring genes in a high-throughput assay for directly monitoring nucleolar rRNA biogenesis (Bryant et al. *Open Biol.* 12:210305) which should be cited and discussed. The authors should prove this alternative interpretation wrong if they wish to maintain their claim.*

Related to the part on GBM, statements like "GBM cell proliferation and tumourigenic potential depend on HEATR1" are void if such a dependence is unspecific. All tumours are dependent on most housekeeping genes. Please qualify the claimed relevance.

We agree with the reviewer in that HEATR1's function is not specifically required in brain tumour initiating cells (TICs). We made this point clearer in Results, describing the decrease in proliferation at later developmental stages of control (non-tumour) type II NSC lineage cells in the *Drosophila* brain (page 9). We now also highlight it in Discussion and cite the recommended publication substantiating HEATR1's role in normal health. In addition, we stress the higher *HEATR1* expression levels detected in brain TICs in the *Drosophila* system even prior to overproliferation onset. Notably, *HEATR1*'s inhibition at this early pre-overproliferative stage prevents *brat* TIC's nucleolar size increase and pointing to higher rRNA generation as detected by lower nascent RNA levels. Yet, it has barely an effect on nucleolar size in normal cell counterparts. The data lead us to propose a higher dependence on HEATR1 by TICs compared to normal cell counterparts, and we believe the revised text explains and provides a better context for our claims (pages 16-17). Of note, a paper just published by Yang and colleagues showing a key role of HEATR1 in ribogenesis during hepatocellular carcinoma growth indicates only limited effects in the proliferation of normal immortalised hepatic cells (Yang *et al*, 2023) (see Discussion page 18).

The manuscript also shows that nucleoli are larger in brat TICs than in control INPs even before TICs start to over-proliferate, which provides indirect, but suggestive data supporting a causative link between HEATR1 and enhanced ribogenesis.

Altogether this is a valuable effort to investigate the mechanisms driving the early stages of TICs. Taking advantage of refined transcriptome analysis from well-defined single-cell types the manuscript reports interesting insight on genes that may play key roles in the earliest stages of brat tumorigenesis. I find this manuscript acceptable for publication once the authors address the few concerns that I have raised.

Other issues

-A FDR < 0.15 is on the low significance side. I would suggest using a more stringent cut-off.

We employed a more stringent FDR of < 0.1 to our transcriptome dataset, a cut-off that still enables detecting KEGG pathway enrichment with FDR < 0.05 despite the small-scale of our single-cell transcriptome approach. We re-did all relevant assays using this more stringent threshold and obtained analogous results, including overrepresentation of proteostasis-associated pathways and OXPHOS metabolism. Respective Figures (Fig 1C-H, Fig EV1D-F) and Datasets EV1-3 were updated accordingly. It is worth noting the successful independent expression validation of identified candidate genes associated with different FDR thresholds as seen in Fig 1C, E and Dataset EV1.

-"MYC expression in nucleoli (Fig 7A-D)" should be "MYC localisation in nucleoli (Fig 7A-D)"

Corrected (page 13).

Please see below *Other remarks* (page 7 of this document)

Reviewer #2:

In this interesting paper, titled "Ribogenesis boost controlled by HEATR1-MYC interplay promotes transition into brain tumour growth", Diaz et al propose a role for increased ribogenesis in promoting growth in brain tumour initiating cells. They utilise the Drosophila larval brain as an in vivo model and patient derived glioblastoma stem cells as an in vitro model.

Their study starts from an impressive experiment whereby single tumour initiating cells are manually picked from the developing Drosophila brain, and their single-cell transcriptomes analysed. They uncover an evolutionary conserved role in brain tumorigenesis for HEATR1, which is found to be upregulated in a tumour background promoting (preceding) growth. HEATR1 is in addition found to

associate with the transcription factor MYC (known to be involved in promoting tumour growth in these contexts), and promote its nucleolar localisation and boosting ribogenesis.

The manuscript and the conclusions are interesting, the experiments are overall well conducted, with different approaches and model systems supporting similar conclusions. The text is well written, and the figures are clearly presented. We are supportive of publishing this manuscript, but several minor concerns remain to be addressed, as listed below.

We thank the reviewer for the comments, acknowledging our study's conclusions as interesting and assays using different models, as well as for the appreciation of our transcriptome analysis of manually harvested single brain cells. We addressed all points raised as detailed below.

1. Figure EV1E requires statistical analysis.

We performed Gene Set Enrichment Analysis (GSEA) of the data, yielding a highly significant normalised enrichment score (NES = 2.31, $p < 1 \times 10^{-7}$) for human orthologue genes differentially expressed in our transcriptome analysis with high glioblastoma stem cell fitness scores as identified by MacLeod *et al.*, 2019. The result supports the value of our transcriptome data. Fig EV1E, corresponding legend and main text were updated.

2. All quantifications of mitosis in Figure 2 and 6 are absolute numbers, without normalisation. PH3 should be normalised to the total number of GFP+ Dpn+ cells, to obtain mitotic index as an indication for proliferation.

The quantifications of mitosis are presented per brain lobe. We now indicate this the Y axis of all respective graphs and apologise for the omission. As requested, we also re-scored all conditions to obtain mitotic indexes. These cannot be calculated per total GFP⁺Dpn⁺ cells because the progeny of *brat*-deficient type II NSC lineages (tumour cells) are Dpn⁺ and such ratios yield values even lower than controls. We thus scored GFP⁺Dpn⁺PH3⁺ per total GFP⁺ cells in all conditions. The results parallel our scores per brain lobe. We present the graphs for mitotic indexes in appendix Figure S1 and S6 and refer to all these in the main text alongside the mitosis scores per brain lobe, as well as in the respective legends for main Figures 2 and 6.

3. All experiments in Drosophila are conducted with HEATR1 loss-of-function, while an overexpression experiment in type II NBs or INPs seems relevant too, to study whether this can promote tumourigenesis. In particular given the clinical phenotype, whereby increased HEATR1 correlates with decreased survival. If a transgenic line is available, this is an obvious experiment that should be included. If not, I feel that generating a new line goes beyond the scope of the revision, and would not change the major conclusions.

We agree with the reviewer in that a *HEATR1* gain of function analysis in *Drosophila* would be valuable, yet indeed there is no transgenic line available to perform the assays and we also felt generating a new line would go beyond the scope and time allocated for revision.

4. If I understand correctly, in the Drosophila work, pntG4>CD8-GFP has been used as a control while the experiments have been conducted with RNAi. Ideally, a non-targeting RNAi from the same library should be included as control. I do not think that all fly experiments need to be repeated, in particular because of the robust phenotypes observed, and the similar observations from different approaches and model systems. However, the findings from the nucleolar analysis in Figure 5P-X are rather subtle, might be related to the different genetic background (control vs KK and TRIP lines used), and would benefit from including an appropriate control.

We now include two additional appropriate controls to the *in vivo* nucleolar analysis performed: one using a non-targeting RNAi from the same library as the *brat*-RNAi strain employed (*Cherry*-RNAi; TRIP collection, Bloomington *Drosophila* Stock Center ID 35785) and a second using the isogenic host

strain for the GD RNAi library (*w¹¹¹⁸*, Vienna *Drosophila* Resource Center, VDRC, ID 60000), the collection that comprises the *HEATR1-RNAi* and *MYC-RNAi* strains used. In both cases, *pntG4* also drives *CD8-GFP*, as in all other conditions tested. We note that the VDRC provides no other controls lines for their GD RNAi collection, and we have not used KK collection lines. The results obtained do not differ from those using the original control. Quantifications are included in Figure 4E and representative images in Appendix Figure S3. Text has been adjusted accordingly, including in Results, Materials and Methods, Figure legends and Appendix sections.

Editorial suggestions:

1. The structure of the manuscript is sometimes hard to follow. Initially, Drosophila work is followed by GBM work, but this is not consistent throughout the manuscript, which can be a bit confusing.

We made rearrangements in Figures 3 to 5 so that *Drosophila* work is also followed by GBM work in this section of the manuscript, making its structure consistent in this way throughout. Necessary rearrangements and order of appearance changes were also performed to Expanded View (EV3-5) and Appendix Supplementary (S4, S5 and S7) figures. Text has been adjusted accordingly, and now includes a brief introductory note on *Drosophila* nucleoli being ultrastructurally less organised than that of mammals but with conserved nucleolar components including FBL (page 10, second paragraph).

2. In the introduction, some parts would better fit in the discussion section eg the first paragraph of page 4.

We would prefer to keep the information on the *brat* model in the Introduction as we believe many colleagues working in tumorigenesis are unfamiliar with this *Drosophila* brain tumour model and it will therefore facilitate their understanding of the results that follow.

3. Figure 2T,U: label the figure referring to 24h ALH.

Labels added (now Figure 2J and K following editorial changes requested by referee #3).

Please see below *Other remarks* (page 7 of this document)

Reviewer #3:

The manuscript by Diaz et al. seeks to understand the gene expression and cellular changes that occur early in cancer development. Based on microarray-based gene expression studies carried out using isolated single INP cells from control and brat mutant brains in Drosophila, the authors focused on the function of the HEATR1 and Myc genes/proteins in regulating cell growth and proliferation. While Myc is long known to regulate rRNA production (and increased cellular biomass), the authors link Myc and HEATR1 to the growth and division of TICs. It is also notable that the changes observed occur prior to the excessive proliferation that is seen when Brat is lost. Overall, this manuscript provides interesting insights using both Drosophila and mammalian systems. I have only minor concerns, as listed below.

We are thankful for the reviewer's assessment and positive comments on our manuscript, namely on our observations prior to *brat* brain tumour overgrowth and interesting insights using both *Drosophila* and mammalian systems. Our response to the minor concerns listed is below.

- Clarification is needed about the single cell microarray studies. The PCA plot implies that only 3 individual nuclei examined for control/brat. Is this correct? And then, were the confirmation RT-PCRs performed on independently isolated nuclei?*

Yes, the reviewer is correct. We performed a small single-cell transcriptome analysis in which the transcriptomes of 3 *brat* TICs and 3 control INPs manually harvested directly from freshly dissected brains were compared. We used Local Pool Error (LPE) for analysis of the data, a method adequate for low sample numbers (Murie & Nadon, 2008). Sample numbers are indicated in the legend of Figure 1A/ workflow, and in Methods. We now added the n numbers also to the main Results' text (page 5). In addition, we expanded the description in Results of the data validation performed using RT-qPCRs (page 6) including that the expression validation of all candidates tested was performed in independent single cell *brat* and control INP cDNA samples (n=3 each) obtained following the same protocol as for the transcriptome analysis.

- *It would have been nice to have a brief justification of why c-Myc was studied in the mammalian system when N-Myc may have been more biologically relevant.*

As suggested, we included a note on MYC family members, highlighting that MYCN (N-Myc) also regulates ribogenesis and is found overexpressed in several neural tumours (page 12). We then describe that *Drosophila* has only one *MYC* gene with highest orthology to *MYC* (*c-MYC*) compared to *MYCN* or other mammalian *MYC* family members and thus sought to focus on it (page 13). The orthology conservation to *MYC* is twice as high as to *MYCN*.

- *In Figure 1H, the String/cytoscape interaction plot shows both mitochondrial and cytoplasmic ribosomal proteins. It wasn't clear to me why mtRps were included here.*

Panel 1H shows *String/cytoscape* interactions between all ribosomal candidates identified in our transcriptome dataset by KEGG pathways analysis (FDR < 0.05; panel 1F). These KEGG pathways categories include both mitochondrial and cytoplasmic ribosomal molecules. We now made this point clear in main text (page 7) and in Fig 1H legend indicating that candidates arising from the KEGG analysis were used for the interaction plot.

- *In figure 6, Drosophila Myc seems to be sometimes in the nucleus and sometimes dispersed in- and outside the nucleus. Is this expected? In this figure, it is extremely difficult to discern any difference between Q' and R'. This (to me) can only be seen in Q' and R'.*

In *Drosophila* brains we observe *MYC* expression predominantly in nuclei but indeed also some signal in the cells' cytoplasm. This pattern is not present in NSC lineage cells that are Deadpan negative (no longer self-renew), which agrees with previous reports showing repression of *MYC* expression by *Brat* in normal NSC lineage progeny (Betschinger *et al*, 2006; Zaytseva *et al*, 2020). We thus have no reason to doubt the detected *MYC* pattern. It is also worth noting that at least in mammals *MYC* has been reported to shuttle in and out of the nuclei, which we mention in the text (page 13). We improved the images on this figure including a new panel 6N with better resolution and arrowheads for better visualisation of the *MYC* expression pattern in the different conditions analysed (Fig 6M-P'; O and P images are now anterior up). To display *MYC* expression in the nucleoli, we now provide higher magnifications (Fig 6Q, R) showing more clearly the small but significant reduction in nucleolar staining upon double *HEATR1*, *brat* inhibition compared to *brat* inhibition alone. We applied a new look-up table (LUT) that better enhances visualisation at this higher magnification (Fig 6Q', R') and re-analysed our *MYC* signal immunostainings using the same quantification method as the one for GBM cells (Fig 7A-C), in which *MYC* signal in nucleoli is normalised here against total (cellular) signal in each cell measured. This is a more robust approach and now consistent between the two assays performed in the different systems. We applied the same LUT to Fig7A''-C'' for consistency.

- *On p13, the authors may want to clarify that Myc binds (weakly) to rDNA in mammalian cells. In Drosophila, as far as I am aware, the activation of rRNA is indirect (Grewal et al. 2005).*

The reviewer is right, and we now highlight this point in Results as suggested (page 14) as well as include it in Discussion (page 18) as follows: In both *Drosophila* and mammals, MYC stimulates rRNA synthesis and ribosome assembly. Yet, unlike in mammals, *Drosophila* rDNA loci lack the consensus E-box MYC binding sites and MYC has been reported not to bind rDNA directly (Grewal *et al*, 2005; Orian *et al*, 2005). Thus, while MYC regulation of ribogenesis seems indirect in *Drosophila*, in mammals it is both indirect and direct via rDNA binding (Arabi *et al*, 2005; Grewal *et al.*, 2005; Orian *et al.*, 2005).

• *In the model presented, Myc and HEATR1 are shown as being regulated independently of each other. The HEATR1 gene has a canonical Myc-binding Ebox in its promoter region (CACGTG). Is it a direct target of Myc?*

The reviewer is right. *HEATR1* contains a canonical MYC-binding E-box in its promoter and has been proposed as a MYC transcriptional target in a genome-wide analysis of MYC-binding sites in Burkitt lymphoma cell lines by chromatin immunoprecipitation followed by sequencing (ChIP-Seq) (Seitz *et al*, 2011), as well as in *Drosophila* S2 cells (Furrer *et al*, 2010; Hulf *et al*, 2005). These studies also reported that *MYC* inhibition decreases *HEATR1* expression. We therefore tested if overexpression of MYC could alter *HEATR1* expression. We observe that MYC overexpression in HeLa cells results in a small but significant upregulation of *HEATR1*. Similarly, in GBM cells, MYC overexpression also leads to an increase in *HEATR1* levels, suggesting *HEATR1* may also be a MYC target in this context. We included these findings in our Results (Fig 7G, right panel and 7H) and Discussion alongside describing the known genetic interaction between MYC and *HEATR1* and postulate that *HEATR1* may also be a MYC target during brain tumorigenesis (pages 14, 18). Our report however demonstrates for the first time a physical binding between MYC and *HEATR1* proteins. Hence MYC and *HEATR1* can interact on protein and transcriptional level. We did not indicate a possible genetic interaction in the proposed model since we did not perform the genetic experiments. We have modified our model (Figure 7J) removing the independent regulation and including only the expression levels changes between normal and brain tumour initiation cells in addition to increased nucleolar MYC localisation in the latter.

Minor:

• *In several figures (at least 2, 5, & 6) panels are referred to out of order in the text.*

We re-arranged figures throughout the manuscript to avoid referring to panels out of order and made corresponding modifications in text. This included re-organising panels in figure 2, previous figure 5 (now figure 4 following editorial changes in response to referee #2) and figure 6.

• *P10 "heatR1" should be "HEATR1"*

Corrected (now in page 11).

• *There seems to be a missing reference in introduction - related to the statement that restoring Brat 24hrs after depletion can prevent tumour growth where it cannot after 48 hours.*

Corrected (page 4). Other references previously at the end of that paragraph are also now better placed.

• *It would be valuable to add gene labels to key genes in the volcano plot shown in figure 1C.*

We labelled *I(2)k09022 (HEATR1)*, the gene explored following our single cell transcriptome analysis.

Please see below *Other remarks*.

Other remarks

In addition to new data added and complying with the reviewer's comments, we have revised our manuscript throughout. The analysis of HEATR1 function was a project spanning several years, which made it difficult to identify all source data. For the results previously presented in Fig 7F and G, we were unable to recover the original data and therefore repeated the experiments. Overall, our new results confirm the original but with some exceptions. We were unable to upregulate 47S pre-rRNA upon MYC transfection in U87MG cells. Due to time constraints and keeping in mind that another group just published a study focusing on HEATR1 (see Discussion, pages 18-19), we decided to focus on HeLa and HEK293 cells in which these assays were possible. Although we no longer could detect an enhancement of the 47s rRNA transcription increase by double transfection of HEATR1 and MYC, a result confirmed in both cell lines (Fig 7F and EV5N), we show that *HEATR1* inhibition can prevent the enhanced 47s rRNA transcription caused by MYC overexpression, supporting a functional interplay between HEATR1 and MYC (Fig 7G). We modify the manuscript text accordingly (see pages 14-15). During this work and following Reviewer #3 comments, we also discovered that MYC overexpression leads to an increase in *HEATR1* levels, corroborating reports proposing HEATR1 as a MYC transcriptional target in other cell contexts and indicating a potential second mode of interplay between the molecules. We added this result to our manuscript (Fig 7G left panel and 7H).

We also note:

For RT-qPCR expression data analysis of candidate gene arising from our single-cell transcriptomics and of respective human orthologue genes (Figs 1E and EV1D), we now applied Holm correction on *p* values following Shapiro-Wilk tests and unpaired 2-tailed *t*-tests to control for family-wise error rate (FWER). The results are similar with all *Drosophila* targets analysed successfully validated and 60% or more of human orthologues found differentially expressed in the sample groups compared.

We re-did our RT-qPCR assays measuring rRNAs upon *HEATR1* inhibition (Fig 5G, H) using cDNA samples generated with random hexamers as the original data used cDNA produced with poly-T primers (see Methods). The results obtained are nevertheless analogous, with reduced rRNA levels observed.

We replaced the representative Fig 2B image as the original was from a different developmental stage, and in Fig 4W we now provide the correct loading control for the NPM1 blot as the original corresponded to another replica sample set of the same assay (see source data).

References

- Arabi A, Wu S, Ridderstrale K, Bierhoff H, Shiue C, Fatyol K, Fahlen S, Hydbring P, Soderberg O, Grummt I *et al* (2005) c-Myc associates with ribosomal DNA and activates RNA polymerase I transcription. *Nat Cell Biol* 7: 303-310
- Betschinger J, Mechtler K, Knoblich JA (2006) Asymmetric segregation of the tumor suppressor brat regulates self-renewal in *Drosophila* neural stem cells. *Cell* 124: 1241-1253
- Furrer M, Balbi M, Albarca-Aguilera M, Gallant M, Herr W, Gallant P (2010) *Drosophila* Myc interacts with host cell factor (dHCF) to activate transcription and control growth. *J Biol Chem* 285: 39623-39636
- Grewal SS, Li L, Orian A, Eisenman RN, Edgar BA (2005) Myc-dependent regulation of ribosomal RNA synthesis during *Drosophila* development. *Nat Cell Biol* 7: 295-302
- Hulf T, Bellosta P, Furrer M, Steiger D, Svensson D, Barbour A, Gallant P (2005) Whole-genome analysis reveals a strong positional bias of conserved dMyc-dependent E-boxes. *Mol Cell Biol* 25: 3401-3410
- Murie C, Nadon R (2008) A correction for estimating error when using the Local Pooled Error Statistical Test. *Bioinformatics* 24: 1735-1736

Orian A, Grewal SS, Knoepfler PS, Edgar BA, Parkhurst SM, Eisenman RN (2005) Genomic binding and transcriptional regulation by the Drosophila Myc and Mnt transcription factors. *Cold Spring Harb Symp Quant Biol* 70: 299-307

Seitz V, Butzhammer P, Hirsch B, Hecht J, Gutgemann I, Ehlers A, Lenze D, Oker E, Sommerfeld A, von der Wall E *et al* (2011) Deep sequencing of MYC DNA-binding sites in Burkitt lymphoma. *PLoS One* 6: e26837

Yang XM, Wang XQ, Hu LP, Feng MX, Zhou YQ, Li DX, Li J, Miao XC, Zhang YL, Yao LL *et al* (2023) Nucleolar HEAT Repeat Containing 1 Up-regulated by the Mechanistic Target of Rapamycin Complex 1 Signaling Promotes Hepatocellular Carcinoma Growth by Dominating Ribosome Biogenesis and Proteome Homeostasis. *Gastroenterology* 165: 629-646

Zaytseva O, Kim NH, Quinn LM (2020) MYC in Brain Development and Cancer. *Int J Mol Sci* 21

Dear Dr. Barros,

Thank you for the submission of your revised manuscript to our editorial offices. I have now received the reports from the three referees that I asked to re-evaluate your study, you will find below. As you will see, the referees now fully support the publication of the study in EMBO reports.

Before I can proceed with formal acceptance, I have these editorial requests I ask you to address in a final revised manuscript:

- I would suggest this slightly modified title:

Ribogenesis boosts controlled by HEATR1-MYC interplay promote transition into brain tumour growth

- Please make sure that the number "n" for how many independent experiments were performed, their nature (biological versus technical replicates), the bars and error bars (e.g. SEM, SD) and the test used to calculate p-values is indicated in the respective figure legends (for main, EV and Appendix figures) of the final revised manuscript. Please also check that all the p-values are explained in the legend, and that these fit to those shown in the figure. Please provide statistical testing where applicable. Please avoid the phrase 'independent experiment', but clearly state if these were biological or technical replicates. Please also indicate (e.g. with n.s.) if testing was performed, but the differences are not significant. In case n=2, please show the data as separate datapoints without error bars and statistics. See also:

<http://www.embopress.org/page/journal/14693178/authorguide#statisticalanalysis>

If n<5, please show single datapoints for diagrams. In particular:

Moreover, I would suggest adding to each legend a 'Data Information' section explaining the statistics used or providing information regarding replicates and scales. See below and:

- Please add scale bars of similar style and thickness to the microscopic images (main and EV figures), using clearly visible black or white bars (depending on the background). Please place these in the lower right corner of the images themselves. Please do not write on or near the bars in the image but define the size in the respective figure legend.

- Please make sure that all figure panels are called out separately and sequentially (main, EV and Appendix figures). Presently, the callout for Fig. 2G should be after 2F, 2I after 2H, 4B after 4A, 4W after 4V, and 6N after 6M. Moreover, these callouts seem missing: Fig. EV4A-J, Appendix Fig. S1A-B, Appendix Fig. S2A-B, Appendix Fig. S3A-B, Appendix Fig. S4A-I, Appendix Fig. S5A-F, Appendix Fig. S6A-B. Moreover, the callouts for Appendix Fig. S7 should be in alphabetical order. Please check.

- Please remove the legends for the Dataset from the manuscript text file and add these as a separate TAB (first TAB) to each Excel file.

- Please remove the referee token from the data availability section and add a direct link to the dataset. Please also make sure the data are public latest on the day of publication of the study.

- Finally, please find attached a word file of the manuscript text (provided by our publisher) with changes we ask you to include in your final manuscript text and comments. Please use the attached file as basis for further revisions and provide your final manuscript file with track changes, in order that we can see any modifications done.

In addition, I would need from you:

- a short, two-sentence summary of the manuscript (not more than 35 words).

- two to four short (!) bullet points highlighting the key findings of your study (two lines each).

- a schematic summary figure that provides a sketch of the major findings (not a data image) in jpeg or tiff format (with the exact width of 550 pixels and a height of not more than 400 pixels) that can be used as a visual synopsis on our website.

Best,

Referee #1:

The authors have conveniently addressed all my concerns.

The only standing issue is the new data regarding the failure to reproduce the upregulation of 47S pre-rRNA upon MYC transfection in U87MG cells and the decision to use HeLa and HEK293 cells as a quick proxi due to the recent publication of a competing study.

Referee #2:

The authors have addressed all our concerns and comments, and we think this is a really interesting manuscript that we would recommend for publication.

Referee #3:

The authors have addressed by concerns.

All editorial and formatting issues were resolved by the authors.

Dr. Claudia Barros
University of Plymouth
School of Medicine and Dentistry
16 Research Way
John Bull Building
Plymouth, Devon PL6 8BU
United Kingdom

Dear Dr. Barros,

I am very pleased to accept your manuscript for publication in the next available issue of EMBO reports. Thank you for your contribution to our journal.

Yours sincerely,
